



# Influence of ENSO on North American subseasonal surface air temperature variability

**Patrick Martineau[1,2], Hisashi Nakamura[1,2], and Yu Kosaka[2]**

[1]Japan Agency for Marine-Earth Science and Technology, Yokohama, Japan
[2]Research Center for Advanced Science and Technology, The University of Tokyo, Tokyo, Japan

**Correspondence:** Patrick Martineau (pmartineau@jamstec.go.jp)

**Abstract.** The wintertime influence of tropical Pacific sea surface temperature (SST) variability on subseasonal variability is revisited by identifying the dominant mode of co-variability between 10–60 d band-pass-filtered surface air temperature (SAT) variability over the North American continent and winter-mean SST over the tropical Pacific. We find that the El Niño–Southern Oscillation (ENSO) explains a dominant fraction of the year-to-year changes in subseasonal SAT variability that are covarying with SST and thus likely more predictable. In agreement with previous studies, we find a tendency for La Niña conditions to enhance the subseasonal SAT variability over western North America. This modulation of subseasonal variability is achieved through interactions between subseasonal eddies and La Niña-related changes in the winter-mean circulation. Specifically, eastward-propagating quasi-stationary eddies over the North Pacific are more efficient in extracting energy from the mean flow through the baroclinic conversion during La Niña. Structural changes of these eddies are crucial to enhance the efficiency of the energy conversion via amplified down-gradient heat fluxes that energize subseasonal eddy thermal anomalies. The enhanced likelihood of cold extremes over western North America is associated with both an increased subseasonal SAT variability and the cold winter-mean response to La Niña.

## 1 Introduction

The El Niño–Southern Oscillation (ENSO), the leading mode of sea surface temperature (SST) variability over the tropical Pacific, has far-reaching impacts on the atmospheric circulation in the Northern Hemisphere through the generation and propagation of a stationary Rossby wave train to the extratropics (Trenberth et al., 1998). This wave train originates in the western subtropical North Pacific, propagates initially northward, and refracts towards North America. The atmospheric response to the warm phase of ENSO (El Niño) is characterized by a strengthening of the North Pacific jet stream that also stretches further eastward, and vice versa for the cool phase (La Niña). It projects strongly on the dominant modes of atmospheric variability over the North Pacific sector (Alexander et al., 2002; Horel and Wallace, 1981), including the Pacific North American (PNA) pattern (Wallace and Gutzler, 1981), under the feedback forcing from the modulated storm track activity (Lau, 1997).

In addition to its effect on the extratropical winter-mean atmospheric circulation, ENSO also influences intraseasonal variability. Nakamura (1996) identified a mode of year-to-year covariability between the winter-mean tropospheric circulation and subseasonal variability over the North Pacific sector. This mode is characterized by extratropical winter-mean circulation anomalies that strongly resemble the atmospheric response to ENSO and the PNA pattern. More specifically, winter-mean anticyclonic anomalies that are associated with the weakened surface Aleutian Low (i.e., negative phase of the PNA) tend to accompany an increase in subseasonal variability over the North Pacific. The associated SST anomalies are characterized by warm anomalies in the central North Pacific, indicative of a possible connection with La Niña. Similarly, Renwick and Wallace (1996) noted an increase in subseasonal variability over the North Pacific in La Niña winters, and Lin and Derome (1997) documented an enhancement of subseasonal variability in negative PNA years.

Since then, many studies have confirmed ENSO's modulations of subseasonal variability (Chen and Van Den Dool, 1999, 1997; Compo et al., 2001; Tam and Lau, 2005). This ENSO influence on subseasonal variability not only affects the mid-tropospheric flow as shown by many of these studies but also has a clear impact on surface air temperature (SAT), potentially modulating the occurrence of weather extremes. In fact, Smith and Sardeshmukh (2000) have shown that the intraseasonal temperature variance is enhanced over the North American West Coast under La Niña conditions.

This ENSO influence on intraseasonal variability may be achieved in part through modulations of the frequency of blocking events, which are prominent and persistent atmospheric circulation anomalies that exert an important influence on SAT variability (Buehler et al., 2011; Martineau et al., 2017; Pfahl and Wernli, 2012; Rex, 1950). For instance, blocking events have been associated with some extreme cold spells in winter (Brunner et al., 2018; Buehler et al., 2011; Cattiaux et al., 2010; Sillmann et al., 2011; Takaya and Nakamura, 2005). Several studies have noted an enhancement of blocking activity during La Niña (Barriopedro and Calvo, 2014; Chen and van den Dool, 1997; Renwick and Wallace, 1996). Others, however, have noted a shift in the preferred location of blocking (Mullen, 1989) or even a decrease in blocking occurrence (Gollan and Greatbatch, 2017; Hinton et al., 2009) during La Niña. These discrepancies likely stem from conceptual differences in the definition of blocking events, which are sometimes defined as a reversal of the zonal flow in the midlatitudes and other times as prominent anticyclonic anomalies. The studies that have defined blocking events as prominent anomalies have reported an increase in the frequency of blocking during La Niña (Barriopedro and Calvo, 2014; Chen and van den Dool, 1997; Renwick and Wallace, 1996), which is in agreement with the aforementioned changes in intraseasonal variability. Considering the link between blocking and weather extremes, this suggests a potential increase in the frequency of extreme cold episodes on subseasonal timescales during La Niña.

Several mechanisms have been proposed to explain ENSO's influence on extratropical subseasonal variability. Tam and Lau (2005) suggested that changes in the tropical source of quasi-stationary Rossby eddies, resulting from ENSO's influence on Madden–Julian Oscillation (MJO) activity, and changes in the propagation of these wave trains under the modulated refractive properties of the midlatitude westerlies by ENSO were among the plausible causes. Changes in barotropic energy conversion, i.e., the direct transfer of kinetic energy between the climatological-mean jet stream and subseasonal eddies, forced by ENSO variability, and changes in the feedback forcing by high-frequency eddies due to shifts in the preferred location of the storm track were also proposed as possible mechanisms (Chen and Van Den Dool, 1999, 1997). High-frequency eddy feedback and interactions between low-frequency variability and the mean flow were also shown to contribute to the enhancement

of subseasonal variability during the negative phase of the PNA (Lin and Derome, 1997) and thus may also be effective during La Niña winters whose extratropical response shares similarities with the PNA as discussed earlier.

Some of these previous studies assumed that the structures of atmospheric circulation anomalies associated with subseasonal variability are predominantly equivalent barotropic, i.e., with slight or even no vertical tilts, and have consequently focused only on barotropic processes to explain ENSO's modulation of subseasonal variability. The role of dynamical processes linked to the vertical dependence of these subseasonal structures, or baroclinicity, in this modulation remains poorly understood. Tam and Lau (2005) nevertheless noted a vertical dependence of the structure of the quasi-stationary waves associated with subseasonal variability over the North Pacific. They have discussed the possibility, without evaluating it though, that baroclinic processes may play a role. Recently, Sung et al. (2019) found that the recent decadal shift of the tropical Pacific into a La Niña-like condition has modified baroclinic energy conversion into the North Pacific Oscillation (NPO; Barnston and Livezey, 1987; Linkin and Nigam, 2008; Wallace and Gutzler, 1981), leading to enhanced monthly-mean temperature extremes over North America. It is thus reasonable to hypothesize that ENSO's influence on subseasonal variability may result, at least in part, from the modulated baroclinicity of the seasonal-mean circulation in the extratropics and the vertical structure of eddies. This hypothesis is plausible since subseasonal anomalies do exhibit vertically tilting structures (Blackmon et al., 1979; Cai et al., 2007; Dole, 1986; Taguchi and Asai, 1987), which play an important role in energizing eddies at this timescale (Cai et al., 2007; Martineau et al., 2020; Sheng and Derome, 1991; Tanaka et al., 2016).

The key goal of this work is thus to assess the role of baroclinic processes in the modulations of subseasonal variability over the North Pacific sector by tropical Pacific variability. As a first step, we perform a singular value decomposition (SVD) analysis to identify the dominant mode of covariability between tropical Pacific SST anomalies and anomalous subseasonal SAT variability affecting the North American continent. By focusing on surface variability instead of mid-tropospheric variability, this study aims to better understand the dynamical processes that regulate persistent subseasonal SAT anomalies that have large socio-economic impacts. Without surprise, ENSO-like SST variability emerges from this analysis as the dominant influence on North American subseasonal SAT variability. In agreement with Smith and Sardeshmukh (2000), La Niña conditions tend to enhance the variability over western North America.

As a second step, we evaluate how changes in the extratropical winter-mean circulation forced by ENSO modulate subseasonal eddy energy. We compare the contributions between baroclinic and barotropic energy conversions from the winter-mean flow to atmospheric circulation anomalies on subseasonal timescales, hereafter referred to as subseasonal

eddies. In addition, we assess the roles of high-frequency eddy feedback and diabatic processes in the energetics. From this analysis, baroclinic energy conversion, which is tied to the vertically tilting structure of subseasonal circulation anomalies, stands out as the primary source of energy by which ENSO modulates subseasonal variability.

## 2 Methodology

### 2.1 Data

This study uses 6-hourly reanalysis data of the global atmosphere from the Japan Meteorological Agency 55-year Reanalysis (JRA-55, Kobayashi et al., 2015) from 1958 to 2019. Variables analyzed include the three-dimensional wind field $(u, v, \omega)$, temperature $(T)$, geopotential height $(Z)$, parameterized diabatic heating $(Q)$, and temperature 2 m above the surface (SAT). SST is obtained from the HadISST dataset (Rayner et al., 2003).

### 2.2 Singular value decomposition analysis

One approach to investigating the influence of ENSO on subseasonal SAT variability is to start from classic ENSO indices (see the next section). The individual indices, however, represent different "flavors" of ENSO that may exert distinct impacts on North American subseasonal SAT variability. Instead of repeating our analysis for all these indices, we apply SVD analysis (Bjornsson and Venegas, 1997; Bretherton et al., 1992) to identify a particular flavor of tropical Pacific SST variability that is optimally related to subseasonal SAT variability over North America. Identifying this optimal influence is not only important to better predict SAT variability from SST anomalies but also contributes to improving the clarity of the rest of our analyses by focusing on the strongest statistical connection.

Here, SVD analysis is used to identify the dominant mode of covariability between winter-mean (December–January–February) SST over the tropical Pacific sector (20° S–20° N, 120° E–70° W) and subseasonal SAT variability (SSV; defined as the local standard deviation of 10–60 d band-pass-filtered 6-hourly SAT during the winter season) over the eastern North Pacific and North American sectors (20–60° N, 140–60° W). The sectors used for the two variables are illustrated in Fig. 2 with dashed rectangles. Results are not sensitive to small variations in these sectors. After obtaining the SST and SSV patterns from the SVD analysis, time series expressing their time evolution ($SVD1_{SST}$ and $SVD1_{SSV}$) are obtained by projecting the original SST and SSV anomaly fields onto these patterns. The SST and SSV patterns shown in this study are not the original patterns directly obtained from the SVD but rather heterogeneous regressions, i.e., SSV regressed onto $SVD1_{SST}$ and SST regressed onto $SVD1_{SSV}$. The heterogeneous patterns are similar to the original homogeneous patterns, while better indicating the coupling between the two fields.

### 2.3 ENSO indices

The $SVD1_{SST}$ time series is compared to classical ENSO indices to identify the index that is optimally related to North American SAT variability. The ENSO indices are obtained by averaging SST anomalies over various sectors followed by a normalization of each index. The sectors are [10–0° S, 90–80° W] for Niño 1 + 2, [5° S–5° N, 150–90° W] for Niño 3, [5° S–5° N, 170–120° W] for Niño 3.4, and [5° S–5° N, 160° E–150° W] for Niño 4 (Bamston et al., 1997).

### 2.4 Energetics of subseasonal eddies

Atmospheric energetics (Lorenz, 1955; Oort, 1964) are used to assess how ENSO modulates the sources of energy sustaining circulation anomalies that produce subseasonal SAT variability (or SSV). Energies and their conversion/generation terms are integrated vertically from the surface to 100 hPa for subseasonal variability that has been extracted by applying a 10–60 d band-pass filter to the 6-hourly data (denoted with primes in the following equations). The basic state, denoted with overbars, is defined as the winter-mean (DJF) fields for individual years that include seasonal-mean anomalies related to ENSO variability.

The eddy available potential energy (EAPE) is defined as

$$\text{EAPE} = \gamma^{-1} \frac{T'^2}{2}, \tag{1}$$

where $\gamma$ is a stability parameter defined as

$$\gamma = \frac{p}{R} \left( \frac{R\hat{\overline{T}}}{C_p p} - \frac{\partial \hat{\overline{T}}}{\partial p} \right). \tag{2}$$

Here $R$ is the gas constant for dry air ($287 \, \text{J} \, \text{K}^{-1} \, \text{kg}^{-1}$), and $C_p$ is the specific heat at constant pressure ($1004 \, \text{J} \, \text{K}^{-1} \, \text{kg}^{-1}$). The stability parameter is here based on temperature averaged over the Northern Hemisphere (denoted by the hat operator). The EAPE is proportional to temperature variance when averaged over a season and receives a strong contribution from the lower troposphere where subseasonal temperature anomalies are strongest (not shown).

Several sources of EAPE are considered. The first is baroclinic energy conversion (CP):

$$\text{CP} = -\gamma^{-1} \left( u'T' \frac{\partial \overline{T}}{\partial x} + v'T' \frac{\partial \overline{T}}{\partial y} \right). \tag{3}$$

It describes how available potential energy is transferred from the winter-mean flow to subseasonal eddies, which is achieved through downgradient eddy heat fluxes. We also

consider feedback forcing on EAPE by high-frequency eddies (Tanaka et al., 2016) that have been extracted with a 10 d high-pass filter (double primes):

$$CP_{HF} = -\gamma^{-1} T' \left( \frac{\partial (u''T'')'}{\partial x} + \frac{\partial (v''T'')'}{\partial y} \right). \tag{4}$$

$CP_{HF}$ describes how high-frequency eddy heat fluxes act to reinforce or dampen subseasonal temperature anomalies.

Diabatic processes can also play a role in the maintenance or dissipation of EAPE. It is evaluated here with

$$CQ = \gamma^{-1} \frac{Q'T'}{C_p}, \tag{5}$$

where $Q$ is the heating rate. Diabatic processes and parameterized vertical heat diffusion, which are provided by JRA-55, are all included in $Q$.

We also investigate ENSO's modulation of eddy kinetic energy (EKE), defined as

$$EKE = \frac{u'^2 + v'^2}{2}, \tag{6}$$

where $u'$ and $v'$ are wind anomalies associated with subseasonal eddies. The sources of EKE considered here include barotropic energy conversion (CK) from the seasonal-mean flow to subseasonal eddies (Oort, 1964; Simmons et al., 1983),

$$CK = \frac{v'^2 - u'^2}{2} \left( \frac{\partial \overline{u}}{\partial x} - \frac{\partial \overline{v}}{\partial y} \right) - u'v' \left( \frac{\partial \overline{u}}{\partial y} + \frac{\partial \overline{v}}{\partial x} \right), \tag{7}$$

the feedback forcing from high-frequency eddies (CK$_{HF}$; Tanaka et al., 2016) defined as

$$CK_{HF} = -u' \left( \frac{\partial (u''u'')'}{\partial x} + \frac{\partial (u''v'')'}{\partial y} \right)$$
$$- v' \left( \frac{\partial (u''v'')'}{\partial x} + \frac{\partial (v''v'')'}{\partial y} \right), \tag{8}$$

and transfers of energy between EAPE and EKE (CPK)

$$CPK = -CKP = -\frac{R\omega'T'}{p}, \tag{9}$$

which is achieved through vertical motion. Here a positive CPK denotes a transfer from EAPE to EKE and vice versa.

Fluxes of energy by the mean flow $(-\nabla \cdot \overline{\mathbf{u}}(EAPE + EKE))$ and pressure work $(-\nabla \cdot (\mathbf{u}'\Phi'))$ are not assessed in this study, since they can basically act to redistribute energy horizontally and thus cannot explain the modulations of EAPE and EKE observed in our analysis. Although there are non-negligible local contributions from these terms, mostly associated with the downstream transport of EAPE and EKE by

the basic-state westerlies over the North Pacific, their overall contribution is small in comparison to CP and CK when integrated over a large domain.

We here evaluate the efficiency of energy conversion by dividing the conversion terms by the total eddy energy (EAPE + EKE) for each winter. For instance, the efficiency of CP may be evaluated as CP$^{eff}$, which is defined as

$$CP^{eff} = \frac{\langle CP \rangle}{\langle EAPE + EKE \rangle}, \tag{10}$$

where the angle brackets denote an average over the months of DJF, a vertical integral from the surface to 100 hPa, and integration over the entire North Pacific (10–87.5° N, 120° E–55° W).

## 2.5 Statistical significance

The statistical significance of linear regressions is assessed using a two-sided $t$ test. Field significance is further controlled following the method described in Wilks (2016) with a false discovery rate control level of $\alpha_{FDR} = 0.1$.

## 3 Results

### 3.1 A preliminary survey of surface air temperature variability

Before investigating modes of covariability, it is useful to look at basic climatological properties of SAT variability. The winter-mean SAT climatology is characterized by stronger SAT gradients over North America in comparison to the surrounding ocean bodies (Fig. 1a). They are especially large in the midlatitudes over the eastern portion of North America but also at high latitudes over the North American West Coast. The northwest–southeast-tilted isotherms reflect the temperature contrast between the warmer North Pacific waters and the colder land surfaces. SSV (Fig. 1b), as defined in this study, is largest over a zonal band stretching over the Bering Strait, Alaska, and western Canada. In contrast, more moderate SSV is found over northern central US, eastern Canada, and part of Greenland. Overall, SSV is markedly larger over land surfaces.

The climatological SSV is then contrasted to climatological high-frequency SAT variability (Fig. 1c), which is associated with the passage of transient synoptic-scale cyclones and anticyclones and has been extracted here by using a 2–8 d band-pass filter. One notices the signatures of storm tracks in high-frequency SAT variability over the North Pacific Ocean and the western boundary of the North Atlantic Ocean. Due to the damping of temperature anomalies by air–sea heat exchanges, however, the maximum high-frequency variability is found over land, specifically over eastern Canada. In that region, high-frequency variability and SSV have similar magnitudes, but elsewhere SSV is markedly dominant. This illustrates that SAT variability at the subseasonal timescale

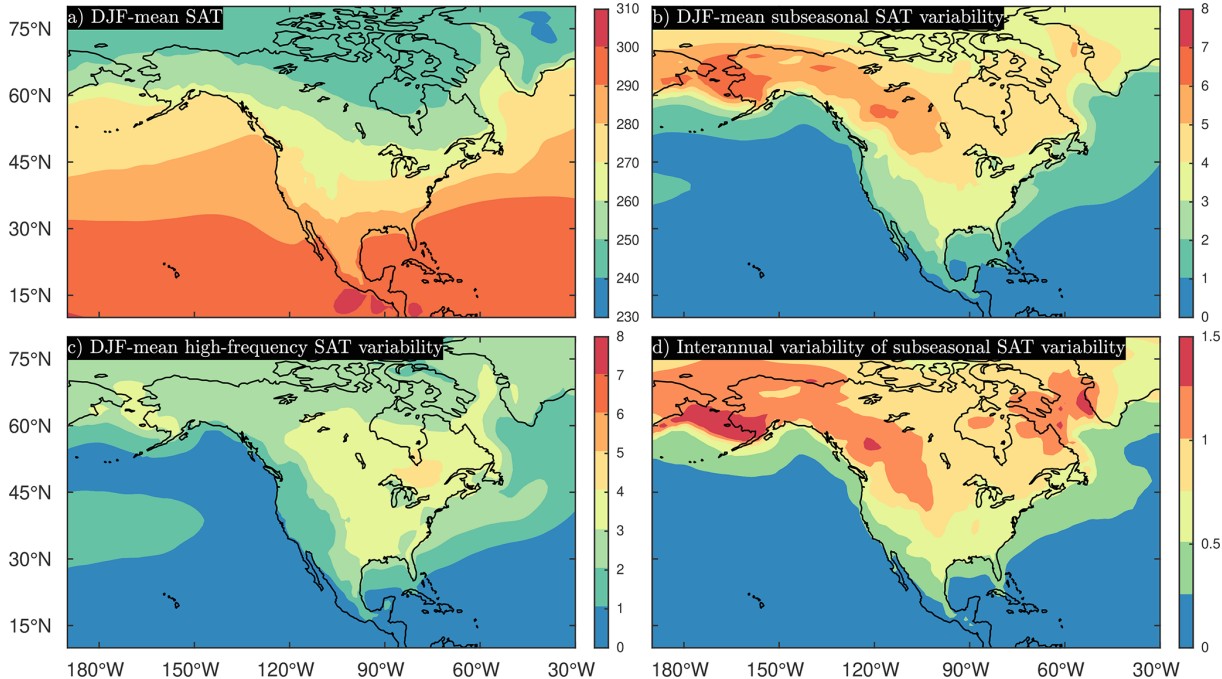

**Figure 1.** Climatological features of SAT variability (1958–2019). The **(a)** DJF-mean SAT, **(b)** DJF-mean subseasonal (10–60 d) SAT variability, or SSV, **(c)** DJF-mean high-frequency (2–8 d) SAT variability, and **(d)** interannual modulations of subseasonal SAT variability are shown with color shadings. All variables are in units of K. Variability is illustrated with the standard deviation.

constitutes an important part of total intraseasonal SAT fluctuations over the North American continent.

The interannual variability of subseasonal SAT variability, calculated as the standard deviation of SSV, is generally large where the climatology of SSV is also large (comparing Fig. 1b and d) with maxima over the Bering Strait, western Canada, northeastern Canada, and southwestern Greenland. It corresponds to fluctuations of about 20 %–30 % of the climatology depending on the specific locations.

## 3.2 Influence of tropical Pacific variability on North American surface temperature variability

The dominant mode of covariability (SVD1) between SSV and SST (Fig. 2), identified through the SVD analysis described in Sect. 2.2, explains about 77 % of the total squared covariance between the two fields. It is characterized by an increase in SSV over the western US, Canada, Alaska, and eastern Siberia, whose magnitude is up to $\sim 10\,\%$ of the SSV climatology (see Fig. 1b). The variability represented by this pattern explains up to about 10 %–20 % of the total local interannual SSV variability. We note that although these anomalies are not significant everywhere according to the statistical test applied, the impact on the vertically integrated energetics that are shown later is more significant. We suspect that variability at the surface is affected by greater noise, or internal variability, which hinders the detection of a statistically significant signal. This enhancement of SSV

is associated with prominent cool SST anomalies over the eastern equatorial Pacific and weaker warm anomalies over the western equatorial Pacific. These significant SST anomalies are strongly reminiscent of the cold phase of ENSO, La Niña. Indeed, the time series representing the temporal variability of this pattern (SVD1$_{\mathrm{SST}}$, Fig. 2c) is strongly anticorrelated to all the four Niño indices (Table 1), which indicates that SVD1$_{\mathrm{SST}}$ essentially reflects SST variability associated with ENSO. SVD1$_{\mathrm{SST}}$ is most strongly anticorrelated to the Niño 3 and Niño 3.4 indices, indicating a dominant link between eastern equatorial Pacific variability and SSV over North America. The anticorrelations between SVD1$_{\mathrm{SSV}}$ and two other Niño indices are also strong and statistically significant. Meanwhile, the correlation between SVD1$_{\mathrm{SST}}$ and SVD1$_{\mathrm{SSV}}$ is significant but rather modest, which suggests that factors other than ENSO, such as internal atmospheric variability or other teleconnections, may also affect subseasonal SAT variability in a similar manner. Such a possibility is briefly explored later.

The regression pattern of winter-mean 500 hPa $Z$ ($Z500$) onto the SVD1$_{\mathrm{SSV}}$ index (Fig. 2b) resembles La Niña's impact on the extratropical atmospheric circulation that has features similar to the negative phase of the PNA. We note, however, that it is more similar to the extratropical response forced by ENSO than the internally generated PNA (Straus and Shukla, 2002). Specifically, cyclonic anomalies are found over the western subtropical Pacific and Canada, while anticyclonic anomalies are over the midlatitude North Pacific

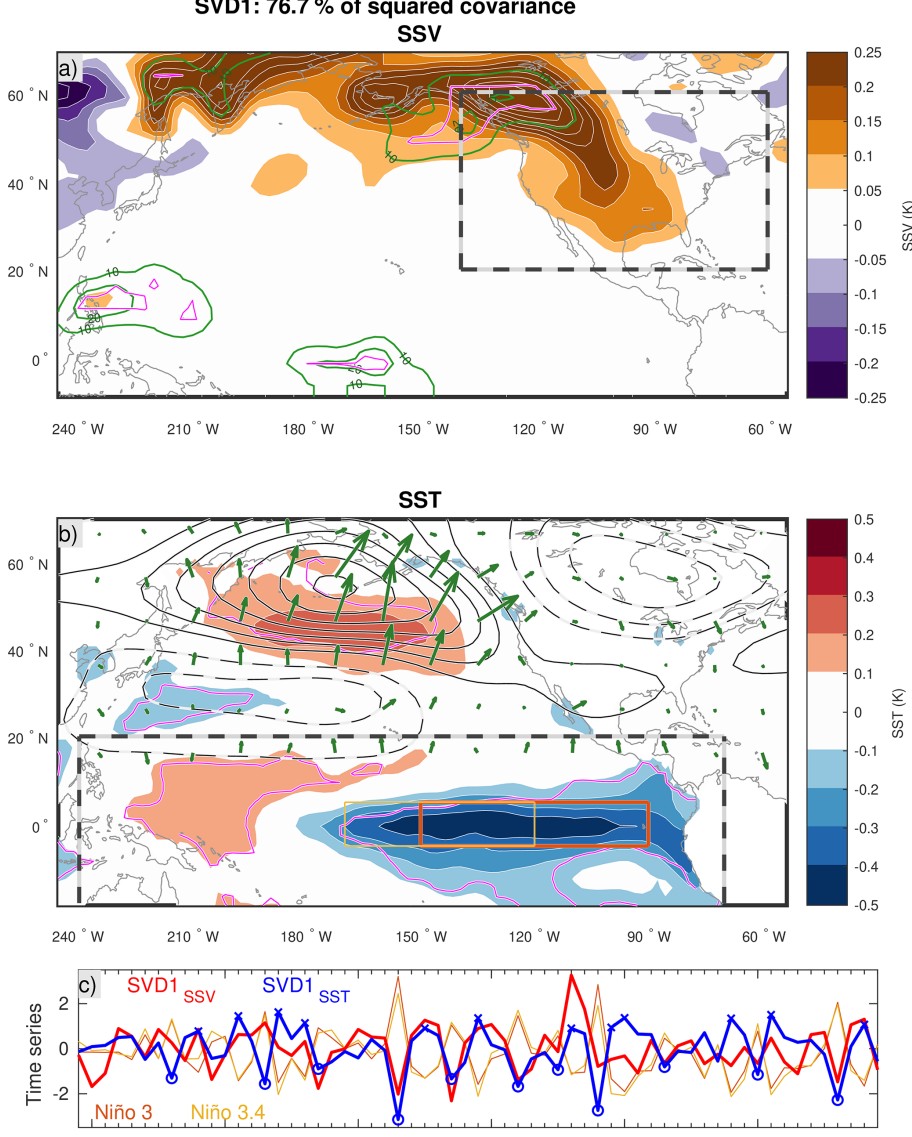

**Figure 2.** Wintertime anomaly patterns of **(a)** subseasonal SAT variability (SSV; 10–60 d) and **(b)** winter-mean SST associated with the dominant mode of covariability (SVD1, corresponding to La Niña) between the two variables (color shadings; anomalies statistically significant at the $\alpha_{\mathrm{FDR}} = 0.1$ significance level are contoured in magenta). The sectors used for the SVD analysis are bounded by bold dashed lines. The green contours superimposed on **(a)** indicate the fraction of interannual SSV variability explained by this mode with 10 % contour intervals (solid and dashed contours are used for positive and negative values, respectively; the 0 line is omitted). $Z500$ anomalies regressed onto SVD1$_{\mathrm{SST}}$ are superimposed on **(b)** with black contours (increment of 5 m, solid and dashed for positive and negative anomalies, respectively). The corresponding wave-activity fluxes (Takaya and Nakamura, 2001) are shown with green arrows. A distance of $1°$ corresponds to a flux of $0.125\,\mathrm{m}^2\,\mathrm{s}^{-2}$. **(c)** Anomaly time series of SSV (SVD1$_{\mathrm{SSV}}$, red) and SST (SVD1$_{\mathrm{SST}}$, blue) associated with SVD1. Two Niño indices are superimposed (domains used to compute these indices are illustrated over the SST pattern in panel **b**).

around 40° N TS1. These anomalies constitute a Rossby wave train refracting around the eastern North Pacific, as suggested by the wave-activity flux (Takaya and Nakamura, 2001). This anomaly pattern accompanies the weakened and more diffluent North Pacific westerly jet in comparison to the climatology. In the warm phase of ENSO (El Niño), by contrast, the continental SSV tends to weaken, and the wave train that comprises the sign-reversed $Z500$ anomalies accompanies the same wave-activity flux as in Fig. 2b.

## 3.3 Processes through which ENSO affects subseasonal eddy activity

In this section, we evaluate ENSO's influence on subseasonal eddy activity by assessing ENSO's modulations of

**Table 1.** Pairwise correlation coefficients between $SVD1_{SSV}$, $SVD1_{SST}$, and the Niño indices (described in Sect. 2.3). Correlations that are significant at the 95 % confidence level are indicated in boldface.

|  | $SVD1_{SST}$ | Niño $1+2$ | Niño 3 | Niño 3.4 | Niño 4 |
|---|---|---|---|---|---|
| $SVD1_{SSV}$ | **0.47** | **−0.43** | **−0.45** | **−0.41** | −0.24 |
| $SVD1_{SST}$ |  | **−0.86** | **−0.98** | **−0.95** | **−0.81** |

various sources/sinks of eddy energy. The rationale is that SSV is produced by weather systems (or eddies) that have deep structures within the troposphere, and thus better understanding of interannual fluctuations of SSV can be acquired through investigating year-to-year changes in processes that energize these eddies. For this analysis, all components of the winter-mean energy budget for subseasonal eddies described in Sect. 2.4 are regressed onto $SVD1_{SST}$. As a reminder, this index is strongly anticorrelated to the Niño 3.4 index. A positive $SVD1_{SST}$ index is, therefore, representative of ENSO's cold phase (La Niña), and all the regression patterns show the linear response to ENSO featuring its cold phase. Note that we have also carried out a composite analysis for El Niño and La Niña winters separately and found salient features to be mostly linear.

Figure 3a–b show anomalies of EAPE and EKE corresponding to a unit standard deviation of the $SVD1_{SST}$ index. Both EAPE and EKE tend to overall increase under La Niña conditions, and the increased EAPE is consistent with the enhanced SSV of SAT over the landmasses. Whereas the EAPE signal is mostly concentrated over landmasses north of the Pacific Ocean and over North America, the EKE signal is particularly large over the subpolar North Pacific. Integrated over the whole North Pacific (Fig. 4), the energy increase is roughly equipartitioned between EAPE and EKE.

Next, we examine the corresponding changes in the conversion of energy from the winter-mean flow to subseasonal eddies through the baroclinic (CP) and barotropic (CK) conversions. A large increase in CP is observed extensively over the subpolar North Pacific with its maximum over the Gulf of Alaska (Fig. 3c). While this could be interpreted as a result of enhanced SSV since the energy conversion is diagnosed from anomalies, an increase in $CP^{eff}$ over the entire North Pacific (Fig. 4) strongly suggests that the stronger CP contributes to the SSV enhancement. The large CP increase occurs where subseasonal eddies exhibit baroclinic structure, especially in the lower troposphere, with climatologically positive correlation between $v'$ and $T'$ (shown later in Fig. 6). Likewise, CK also tends to increase over the midlatitude North Pacific (Fig. 3d), but overall, the contribution of CK is smaller than that of CP (Fig. 4).

As per their definition, CP and CK depend on both the winter-mean flow configuration and eddy properties. To assess their relative importance, we compute composite differences between winters when the normalized $SVD1_{SST}$ is above 0.75 or below −0.75, as the 12 and 11 winters indicated with crosses and open circles, respectively, in Fig. 2c. The composite differences in which both the eddy properties and basic-state properties are allowed to vary from year to year (Fig. 5e–f) are contrasted to the corresponding composites in which the basic-state properties (Fig. 5a–b) or eddy properties (Fig. 5c–d) are fixed to their climatologies from 1958 to 2019. Statistical significance is assessed through a bootstrapping approach with randomly resampled (3000 times) composites of the same sample size as those shown in Fig. 5. The comparison reveals that year-to-year changes in eddy properties are essential to explain the enhanced CP over the Pacific sector (comparing Fig. 5a to 5e). Although the total composite difference in CP and the one using constant eddy properties are both significant over the northern Pacific (Fig. 5c, e), the significance is somewhat reduced for the composite difference with the constant basic state (Fig. 5a). We suspect this may be due to a cancellation of ENSO-unrelated noise when fluctuations in both eddy and winter-mean properties are considered, which contributes to increasing the statistical significance. It may also hint that modulations of eddy structure tend be coherent with changes in the winter-mean flow. Over the domain of enhanced CP, we find evidence for a stronger positive correlation between $v'$ and $T'$ (Fig. 6) throughout the depth of the troposphere, which indicates that the structure of subseasonal eddies is more adequate in La Niña winters to extract energy for their baroclinic growth from the meridional thermal gradient associated with the Pacific jet. The increased CP is also found to arise from the tendency for the climatologically positive and negative correlations between $u'$ and $T'$ to be enhanced over Alaska and the Sea of Okhotsk, where the zonal temperature gradients are climatologically positive and negative, respectively. The temperature gradients associated with planetary waves tend to strengthen in the La Niña winters (c.f., Fig. 2b). Meanwhile, the enhanced barotropic conversion (CK) over Alaska results from changes in eddy properties, while the changes over the western North Pacific appear to result from a combination of changes in both eddy and winter-mean flow properties (Fig. 5 right).

The contributions of CP and CK are found to be much larger than the feedback forcing by high-frequency transients ($CP_{HF}$ and $CK_{HF}$), which are weaker and contribute minimally to the changes in the energetics (Figs. 3e–f and 4). Similarly, the diabatic feedback (CQ) has a negligible contribution.

Finally, we also investigate the transfer (CPK) between EAPE and EKE. We find that CPK is enhanced over a broad domain stretching northeastward from the western subtropical North Pacific to Alaska (Fig. 3h). This domain is collocated with the region of enhanced CP (Fig. 3c), which indicates that an important portion of the gains in EAPE through CP is transferred to EKE in situ. This transfer is small compared to CP but about half of CK and thus relevant to the observed increase in EKE. The correlation between $\omega'$

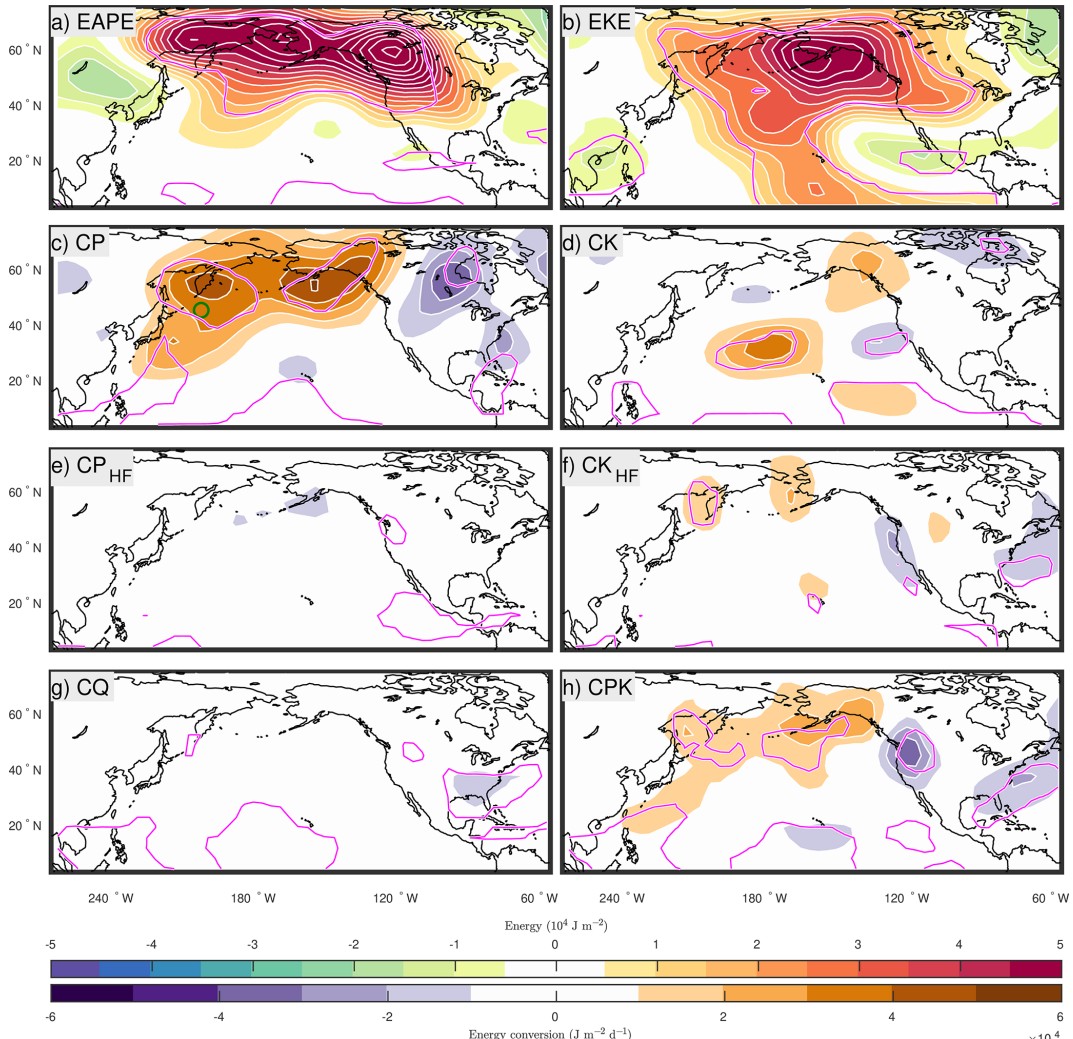

**Figure 3.** (a) Eddy available potential energy (EAPE), (b) eddy kinetic energy (EKE) of subseasonal eddies and energy sources/sinks (c–h), all regressed onto the $SVD1_{SST}$ index. (c) CP and (d) CK are the baroclinic and barotropic energy conversions, respectively. (e) $CP_{HF}$ and (f) $CK_{HF}$ represent the baroclinic and barotropic high-frequency eddy feedbacks. (g) CQ is the diabatic feedback. (h) CPK represents transfers from EAPE to EKE. Correlations that are statistically significant at the $\alpha_{FDR} = 0.1$ significance level are contoured in magenta. The green circle in panel (c) indicates the reference location for Fig. 8.

and $T'$ over the western North Pacific is slightly decreased in $SVD1_{SST} > 0.75$ (Fig. 6), which suggests that changes in CPK are partly due to changes in the structure of eddies that increase their efficiency in transferring energy from EAPE to EKE.

### 3.4 Changes in propagation and structure of subseasonal eddies

In this section, we assess ENSO's influence on the propagation of wave activity in relation to the structure of subseasonal eddies. Subseasonal eddy propagation is first assessed by using the wave-activity flux for stationary Rossby waves (Takaya and Nakamura, 1997, 2001) computed from 10–60 d band-pass-filtered 6-hourly $Z300$. It is computed

for each time step after filtering and then averaged over the winter months (DJF) before being regressed onto $SVD1_{SST}$. The flux, which is climatologically eastward (not shown) reflecting the eastward group velocity of stationary Rossby waves, tends to be enhanced during La Niña winters (Fig. 7). The enhanced eastward flux maximizes over the subpolar North Pacific, where CP is enhanced, and over western North America, where SSV also increases noticeably. The eastward wave-activity flux is also enhanced just east of the region of enhanced CK over the subtropical North Pacific.

To better understand how quasi-stationary eddies can extract energy from the winter-mean flow through their heat fluxes more efficiently in La Niña winters than in El Niño winters, we construct one-point regression maps to highlight the vertical structure of these eddies separately for those

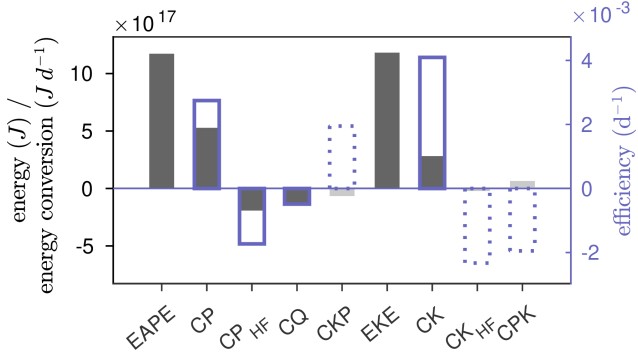

**Figure 4.** The energetics of subseasonal eddies are integrated over the North Pacific (10–87.5° N, 120° E–55° W) and regressed onto SVD1$_{\text{SST}}$. Raw changes in energy and sources/sinks are shown with solid bars. Significant values are shown with a darker shade of gray. Blue boxes indicate changes in the efficiency with significant changes indicated with solid borders. Quantities shown include eddy available potential energy (EAPE), baroclinic energy conversion (CP), high-frequency eddy baroclinic feedback (CP$_{\text{HF}}$), diabatic feedback (CQ), transfers from EKE to EAPE (CKP), eddy kinetic energy (EKE), barotropic energy conversion (CK), high-frequency barotropic feedback (CK$_{\text{HF}}$), and transfers from EAPE to EKE (CPK).

two types of winters, shown in zonal sections in Fig. 8. All the time series have been exposed to a 10–60 d band-pass filter before evaluating the regressions. The reference time series are normalized so that the regressed patterns represent circulation anomalies associated with typical SAT variability. The eddy structures are evaluated over the midlatitude North Pacific, where the enhanced positive correlation between $v'$ and $T'$ (Fig. 6) hints to important zonal structural changes that lead to a substantial modulation of CP (Fig. 3c). For a robust illustration of the structure of eddies over that sector, it is preferable to use a local reference grid point in this analysis (as indicated with a green circle in Figs. 3c and 6). First, we note that the retrieved structure of the anomalies is clearly baroclinic, with westward-tilting geopotential height anomalies and eastward-tilting temperature anomalies. Such structures are known to accompany net poleward heat fluxes, allowing the eddies to extract available potential energy from the basic-state flow. The net poleward heat flux associated with these structures (Fig. 8c–d) is larger for SVD1$_{\text{SST}}$>0.75, due to an enhancement of poleward heat transport to the west of the reference longitude, while southward transport to the east is mostly the same. Although rather subtle, the changes in the vertical structure of subseasonal eddies, manifested also as the enhanced positive $v'$–$T'$ correlation (Fig. 6), contribute to a more efficient downgradient heat transport during La Niña winters (SVD1$_{\text{SST}}$>0.75), thus leading to more efficient extraction of energy from the winter-mean flow through baroclinic conversion (Fig. 4).

Next, we perform lag-regression analysis to identify the typical structure of quasi-stationary eddies that are associated

with subseasonal SAT variability over western North America (Fig. 9). The analysis is carried out with reference subseasonal SAT time series over Alaska and Colorado. These locations are chosen because SVD1 indicates a large impact on SSV over these sectors (Fig. 2). The lag regression maps corresponding to Alaska SAT variability show a clear wave train developing downstream, in agreement with eastward wave-activity fluxes (Fig. 9a, c, e). The wave train is mostly stationary, although slow retrogression is hinted for the North Pacific anomaly from lag −3 (day) to lag 0 (Fig. 9c, e), as typically observed in this maritime region (Branstator, 1987; Kushnir, 1987; Nishii et al., 2010). At all lags, positive correlations are observed over the subtropical northwestern Pacific, suggesting a potential subtropical origin to this wave train. Precursors are also observed over Russia at lag −6 (Fig. 9a) as well as lags −9 and −12 (not shown), indicating that this wave train may also originate from the extratropics.

The corresponding lag regression maps for a reference SAT index over Colorado also show an eastward-developing wave train (Fig. 9f). The origin of this wave train can be traced back in part to the subpolar northwestern Pacific and the subtropical central Pacific (Fig. 9b, d). In this case, the phase of the wave train is seen to move slowly eastward especially around the Rockies (Hsu and Wallace, 1985). As illustrated in Fig. 9, atmospheric circulation patterns associated with localized SAT variability are quite sensitive to the reference location. These patterns share similar features, such as their spatial scale and meandering, with the circulation anomalies associated with the leading modes of SAT variability (Lin, 2015). The exact location of their cyclonic and anticyclonic centers of action are, however, not the same. They may correspond to modes of lesser importance or combinations of the leading modes. Other reference locations over North America were assessed and revealed different circulation anomalies (not shown).

The two quasi-stationary wave trains revealed by the regression analysis propagate through the North Pacific sector where CP and CK are enhanced. Significant differences in the amplitude of these wave trains are observed between the two phases of SVD1$_{\text{SST}}$. For the typical wave train affecting Alaska, differences in the $Z500$ regression pattern (Fig. 8) are positive over Alaska, where the $Z500$ anomalies are typically positive and are negative over the cyclonic anomalies upstream (Fig. 9a–b). This indicates an overall intensification of the wave trains under La Niña conditions. It is consistent with the enhanced eddy energy and wave-activity fluxes discussed previously. Amplification is not as clear for the typical wave train affecting Colorado. Positive amplitude differences are nevertheless found for the anticyclonic anomalies located upstream.

## 3.5 Impact on extreme temperature events

The impact of SVD1$_{\text{SST}}$/ENSO variability on the occurrence of persistent weather extremes is now investigated. Cold

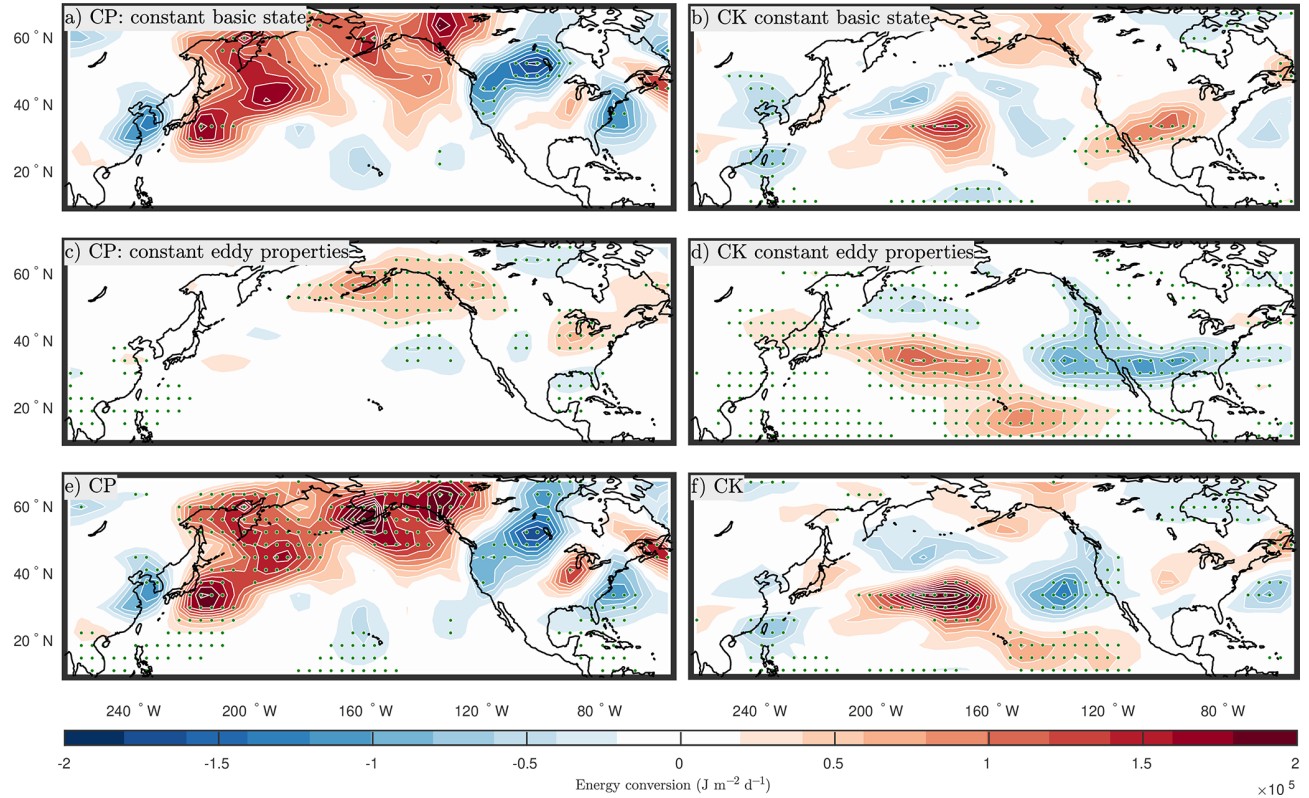

**Figure 5.** Baroclinic energy conversion (CP; **a, c, e**) and barotropic energy conversion (CK; **b, d, f**) composite differences between the positive (SVD1$_{SST}$>0.75) and negative (SVD1$_{SST}$< − 0.75) phases of SVD1 (representing La Niña) are calculated by keeping either the background flow properties (**a, b**) or eddy properties (**c, d**) constant every year. For reference, the composite difference when both eddy properties and the basic state can change from year to year is shown in panels (**e**) and (**f**). Composite differences that are significant at the $\alpha_{FDR} = 0.1$ significance level are dotted.

(warm) extreme days are defined as the days when the 10 d low-pass-filtered SAT anomaly falls below (rises above) the 5th (95th) percentiles at each grid point over the 62 winters. Their frequency, calculated as the percentage of winter days each year, is then regressed onto the SVD1$_{SST}$ time series. The spatial patterns of changes in the frequency of weather extremes (Fig. 10a–b) are similar to the winter-mean SAT response (Fig. 10c) with enhanced frequency of cold extremes over the regions that are colder than normal and vice versa. Generally, the corresponding relationship holds also between warm extremes and winter-mean SAT. This indicates that the winter-mean response to ENSO variability is related to the frequency of extremes, through shifts in the probability distributions of temperature. This is clearly observed, for instance, over Manitoba (Fig. 11d), where the whole probability density is shifted towards colder temperatures for SVD1$_{SST}$>0.75, while the standard deviation is almost identical.

However, an important mismatch is observed in the response of warm and cold extremes over western and southern North America (Fig. 10a–b). In these sectors, increases in the frequency of cold extremes are not matched with similar de-

creases in the frequency of warm extremes and vice versa. In northern BC (Fig. 11a) and western Canada, for example, colder winter-mean SAT under the La Niña conditions accompany an increase in the frequency of cold extremes, while the corresponding changes in the frequency of warm extremes are rather small and insignificant over land. In this case, an increase in variability broadens the distribution towards cold temperatures. Thus the likelihood of cold extremes is augmented, while warm extremes mostly maintain a similar frequency. On the opposite, in Texas (Fig. 11c), increased variability broadens the distribution towards warmer temperatures, which accompanies the increased frequency of warm extremes but little change in the frequency of cold extremes. Some other sectors, such as Colorado (Fig. 11b), undergo virtually no shift in the winter mean. There, a small increase in the frequency of both cold and warm extremes is due to an increase in subseasonal variability that broadens the temperature distribution towards both warm and cold temperatures.

Over TS4 some sectors, changes in skewness are also associated with the frequency of warm and cold extremes, but their contribution is overall less organized spatially and sta-

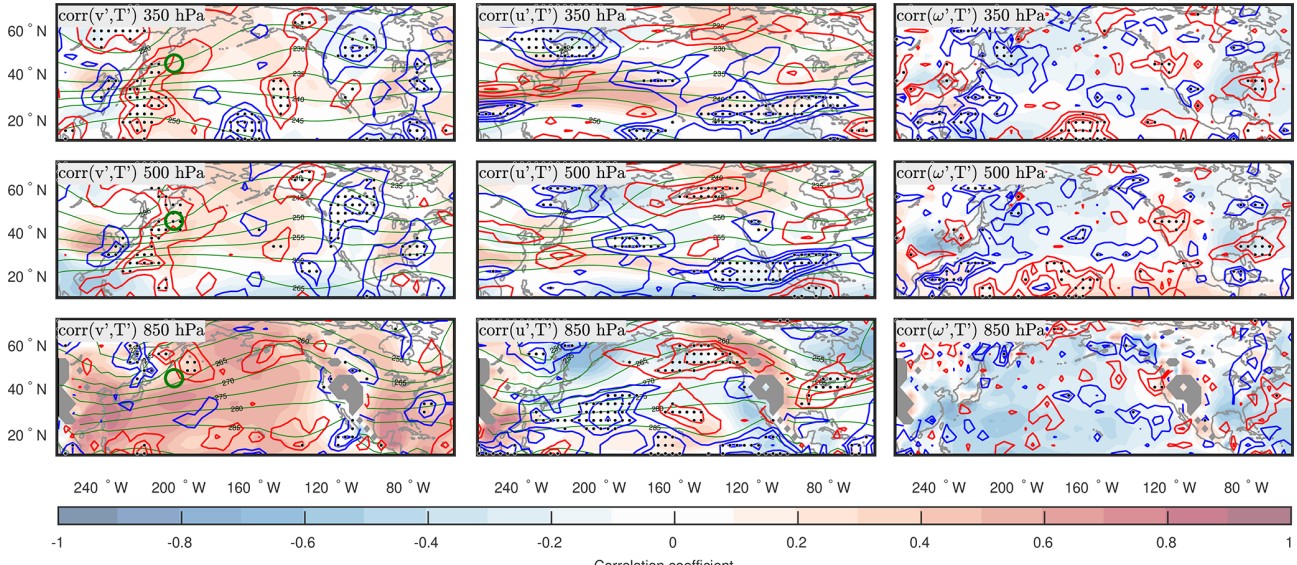

**Figure 6.** Maps of climatological correlation coefficients (color shading) between $v'$ and $T'$ (left), between $u'$ and $T'$ (center), and between $\omega'$ and $T'$ (right). The respective composite differences between winters of $SVD1_{SST} > 0.75$ and $SVD1_{SST} < -0.75$ are superimposed with blue and red contours with an interval of 0.1 for negative and positive differences, respectively. For reference, the temperature climatology is shown with green lines. Composite differences that are significant at the $\alpha_{FDR} = 0.1$ significance level are dotted. The green circle in the left panels indicates the reference location for Fig. 8.

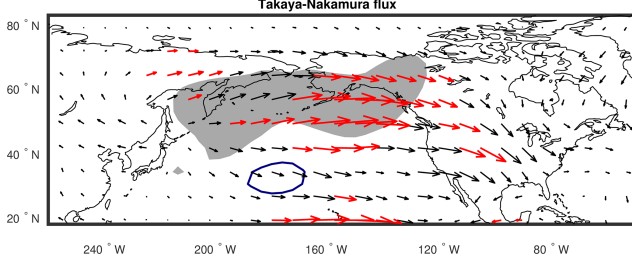

**Figure 7.** The wave-activity flux (Takaya and Nakamura, 2001) of 10–60 d band-pass-filtered eddies at 300 hPa regressed onto the $SVD1_{SST}$ time series. The anomalous flux whose meridional or vertical component is significant at the $\alpha_{FDR} = 0.1$ significance level, assessed with a $t$ test on the correlation coefficients, is shown in red. A distance of $1°$ corresponds to a flux of $0.67\,\mathrm{m^2\,s^{-2}}$. Regions where CP and CK are larger than $3 \times 10^4\,\mathrm{J\,m^{-2}\,d^{-1}}$ are denoted with gray shading and a blue contour, respectively (see Fig. 2c–d).

tistically insignificant (not shown). One key message is that it is not possible to infer changes in the frequency of extremes from changes in subseasonal variability alone. One needs to carefully examine the detailed changes in the probability density of temperatures including shifts towards colder or warmer temperatures.

## 3.6 Modulation of subseasonal SAT variability unrelated to ENSO

As mentioned earlier, $SVD1_{SSV}$ is correlated rather moderately with $SVD1_{SST}$ and the Niño indices (Table 1), which suggests that a substantial fraction of year-to-year variations in SSV over the North American West Coast may arise from internal atmospheric variability. The processes responsible for this variability are briefly assessed. The component of $SVD1_{SSV}$ that is uncorrelated with $SVD1_{SST}$ ($SVD1R_{SSV}$) is first identified as the residual of the linear regression between the two indices. By regressing SSV onto the index (Fig. 12a), we find an amplification of SSV whose spatial structure is similar to the one previously identified (Fig. 2a) and whose intensity is notably augmented over North America (up to $\sim 15\,\%$ of the climatology (Fig. 1b) and $50\,\%$ of the total interannual variability explained). The correlations of $SVD1R_{SSV}$ with the ENSO indices are indeed quite weak and overall insignificant (Table 2). The corresponding grid-by-grid correlation with SST is significant only in the central equatorial Pacific (Fig. 12b). Stronger SST anomalies are nevertheless found in the midlatitude North Pacific with a pattern somewhat reminiscent of the North Pacific Gyre Oscillation (NPGO) (Di Lorenzo et al., 2008). While the winter-mean circulation pattern associated with $SVD1R_{SSV}$ (Fig. 12b) shares some similarities with the negative phase of the PNA, it is also similar to the negative phase of the NPO. In fact, Di Lorenzo et al. (2008) indicated that the NPGO is driven by wind stress curl anomalies associated with the

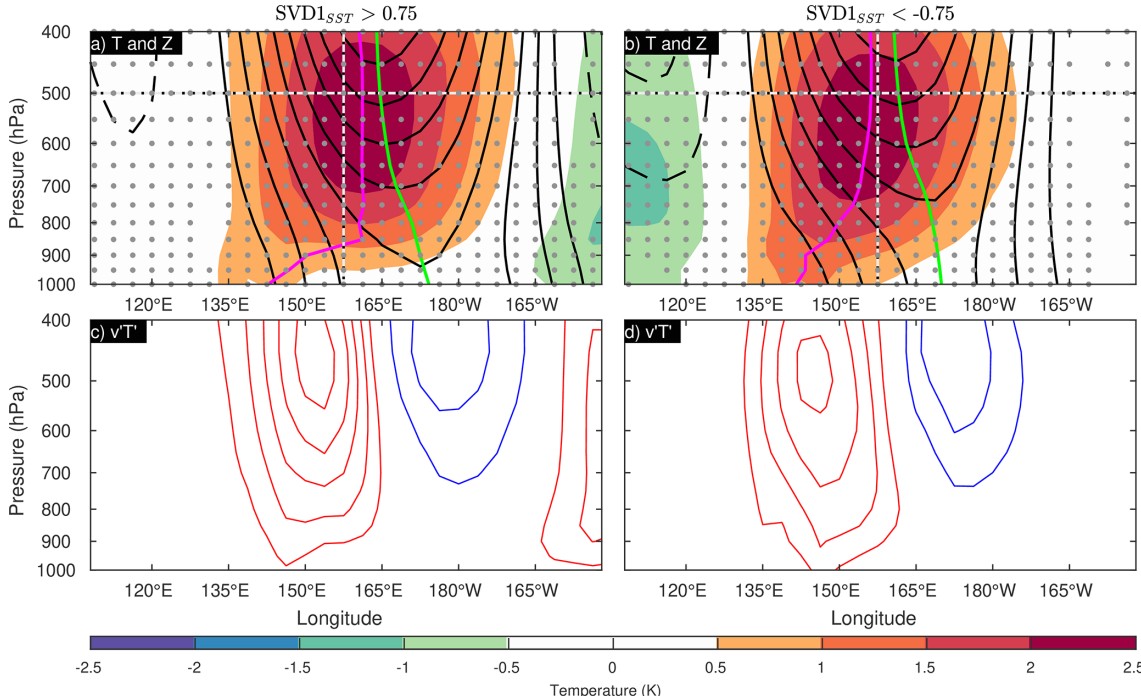

**Figure 8. (a, b)** Zonal sections of subseasonal anomalies in geopotential height (black contours with intervals of 10 m; solid and dashed lines are used for positive and negative values, respectively) and temperature (shadings) both regressed onto the reference time series of 500 hPa geopotential height at [45° N, 157.5° E] (green circles on Figs. 3c and 6), for winters when (**a**; left) SVD1$_{SST}$ >0.75 and (**b**, right) SVD1$_{SST}$ < −0.75. The maxima of the height and temperature anomalies at individual pressure levels are connected vertically with green and magenta lines, respectively. (**c, d**) The associated meridional heat fluxes ($v'T'$) are contoured at intervals of 1 m K s$^{-1}$ with red and blue lines for positive (northward) and negative (southward) values, respectively. Stippling indicates geopotential height values that are statistically significant at the $\alpha_{FDR} = 0.1$ significance level.

**Table 2.** Pairwise correlation coefficients between SVD1R$_{SSV}$ and the Niño indices (described in Sect. 2.3). None of the correlations are significant.

|                | Niño 1 + 2 | Niño 3 | Niño 3.4 | Niño 4 |
|----------------|------------|--------|----------|--------|
| SVD1R$_{SSV}$  | −0.03      | 0.02   | 0.05     | 0.17   |

NPO. We argue that SVD1R$_{SSV}$-associated variability is overall related to the internal variability of the eddy-driven jet over the North Pacific. The feedback of NPGO-like SST anomalies onto the atmospheric anomalies in Fig. 12b needs to be addressed in future studies. Like SVD1, SVD1R$_{SSV}$ enhances the efficiency of the baroclinic conversion of energy from the winter-mean flow over the North Pacific and the propagation of quasi-stationary waves towards North America (not shown), which acts to enhance SAT variability over the continent.

## 4 Summary

By identifying the dominant mode of interannual covariability between winter-mean tropical SST and subseasonal SAT

variability, this study confirms the prominent role of ENSO in modulating the SAT variability over North America. El Niño and La Niña tend to reduce and enhance the SAT variability, respectively (Smith and Sardeshmukh, 2000). Among the classical ENSO indices, the Niño 3 and Niño 3.4 indices are most closely correlated with the mode of variability identified in this work. This dominant mode explains about 77 % of the squared interannual covariance between SST and subseasonal SAT variability and a more modest fraction of variance (up to ∼ 10 %–20 % in some sectors) of the total subseasonal SAT variability including both SST-forced and internal components. Although rather small, this fraction is nonetheless important because it represents what is predictable from SST variability, unlike atmospheric internal variability that is less predictable.

Energetics of subseasonal atmospheric eddies reveal that La Niña is not only accompanied by an augmentation of EKE (Chen and Van Den Dool, 1999) but also by an increase in EAPE over the North Pacific sector, which is consistent with the rise of subseasonal SAT variability. In fact, the modulation of baroclinic energy conversion by ENSO is found more important than the barotropic processes emphasized in previous studies (Chen and van den Dool, 1997; Tam and Lau,

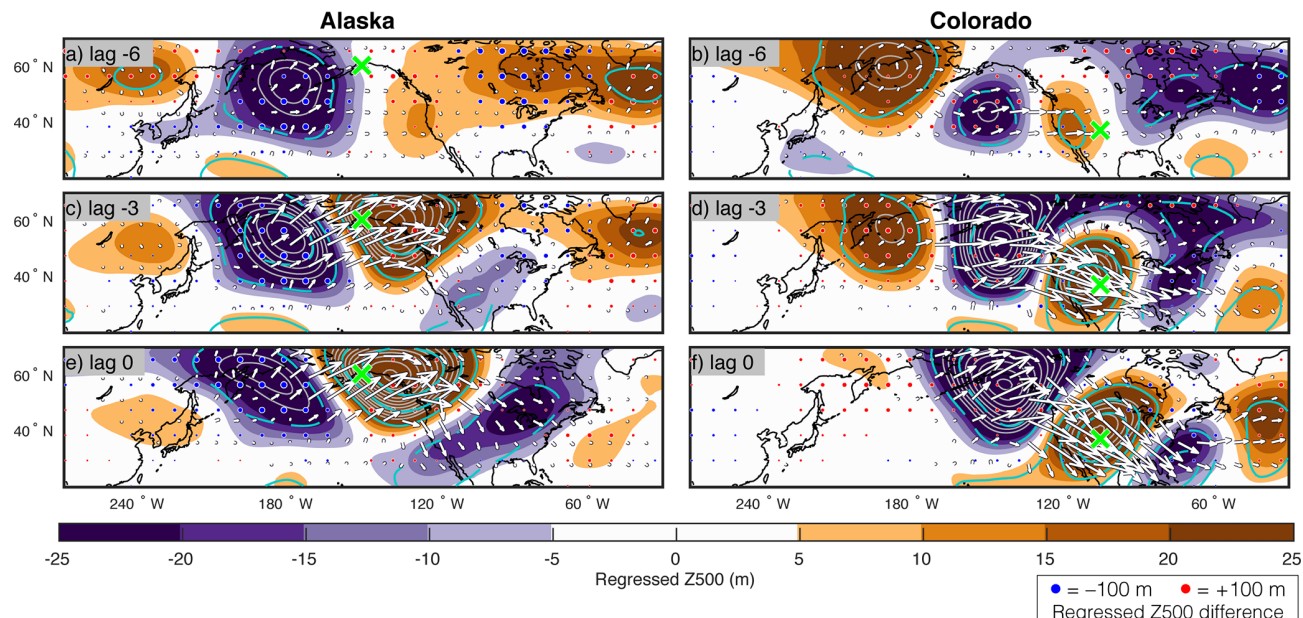

**Figure 9.** One-point regression maps of Z500 anomalies (color shading) with respect to reference SAT time series over **(a, c, e)** Alaska (61° N, 150° W) and **(b, d, f)** Colorado (39° N, 105° W). The corresponding correlation is superimposed with cyan contours (increments of 0.2). The regression is performed separately for years when SVD1$_{SST}$ ≥ 0.75 and SVD1$_{SST}$ ≤ −0.75 TS2 (indicated with blue crosses and circles in Fig. 1, respectively), and the average of the two patterns is shown in this figure. The reference SAT time series for the positive and negative phases are normalized independently before carrying out the regression. Differences between the SVD1$_{SST}$ ≥ 0.75 and SVD1$_{SST}$ ≤ −0.75 TS3 patterns that are significant at the $\alpha_{FDR} = 0.1$ significance level are shown with blue dots for negative differences and red dots for positive differences. The dots are scaled according to the magnitude of the difference. For reference, the dots illustrated in the legend represent a difference of 100 m. The wave-activity flux (Takaya and Nakamura, 2001) evaluated with the regressed Z500 anomalies is shown with white arrows with a distance of 1° corresponding to a flux of 2/3 m² s⁻².

2005). The baroclinic energy conversion to subseasonal eddies is achieved through their heat fluxes that are downgradient of the winter-mean temperature associated with the Pacific jet. Alternatively, this conversion can be interpreted as the anomalous thermal advection by subseasonal eddies acting on the climatological temperature gradient in such a way that it reinforces eddy temperature anomalies. In contrast to the baroclinic energy conversion, the net feedback forcing from high-frequency eddies migrating along the Pacific storm track, which was suggested as an important process (Chen and Van Den Dool, 1999, 1997), is much smaller. It is explained by the fact that previous studies have only assessed the budget of kinetic energy in the upper troposphere, which overemphasizes the feedback from high-frequency eddies on the kinetic energy (Lau and Nath, 1991) and overlooks the cancellation between high-frequency eddy feedbacks onto the eddy available potential energy and eddy kinetic energy (Tanaka et al., 2016).

Although lag-regression maps suggest that the subseasonal eddies affecting SAT over North America originate in part from the tropics, the modulation of subseasonal eddy energetics by ENSO is dominated by modulated baroclinic energy conversions in the mid- to high latitudes. This suggests that ENSO's influence is not a simple manifestation of modu-

lated tropical sources of subseasonal variability, as suggested by Tam and Lau (2005). This conclusion is also supported by the absence of significant changes in EAPE or EKE in the western tropical Pacific and the fact that the modulation of energy sources by diabatic processes is comparatively very small. It is more likely that subseasonal wave trains forced by normal levels of tropical convective activity can extract more energy from the winter-mean Pacific jet under La Niña conditions as they propagate through the mid- and high latitudes.

We have further revealed that changes in the properties of subseasonal eddies are essential for the enhancement of baroclinic energy conversion during La Niña winters. In comparison, ENSO-related changes in winter-mean flow properties have a rather modest direct impact on the energetics. The background flow properties, however, may have an indirect impact through their influence on the propagation and structure of subseasonal eddies. Over the midlatitude North Pacific, subseasonal anomalies in eddy velocity and temperature are overall better correlated during La Niña winters, which translates into larger downgradient eddy heat fluxes and consequently into more efficient baroclinic energy conversion for their growth and maintenance. The enhanced correlation results from a more pronounced vertical tilt of eddies

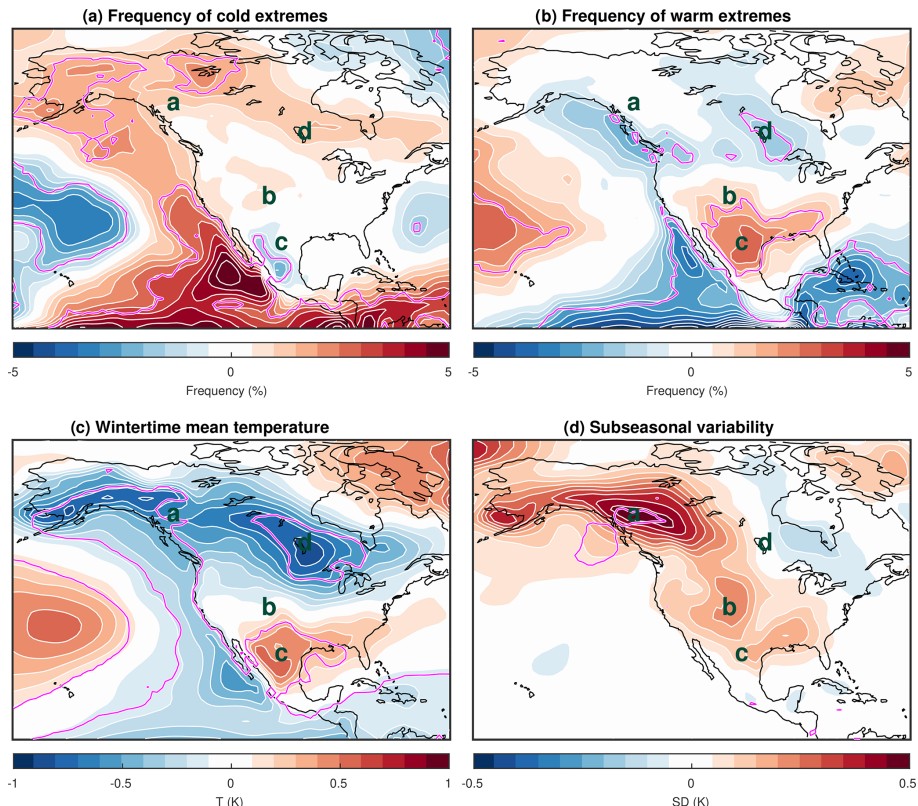

**Figure 10.** Regression of **(a)** frequency of cold extremes, **(b)** frequency of warm extremes, **(c)** winter-mean SAT, and **(d)** subseasonal SAT variability onto the SVD1$_{SST}$ index. Values that are statistically significant at the $\alpha_{FDR} = 0.1$ significance level are contoured in magenta.

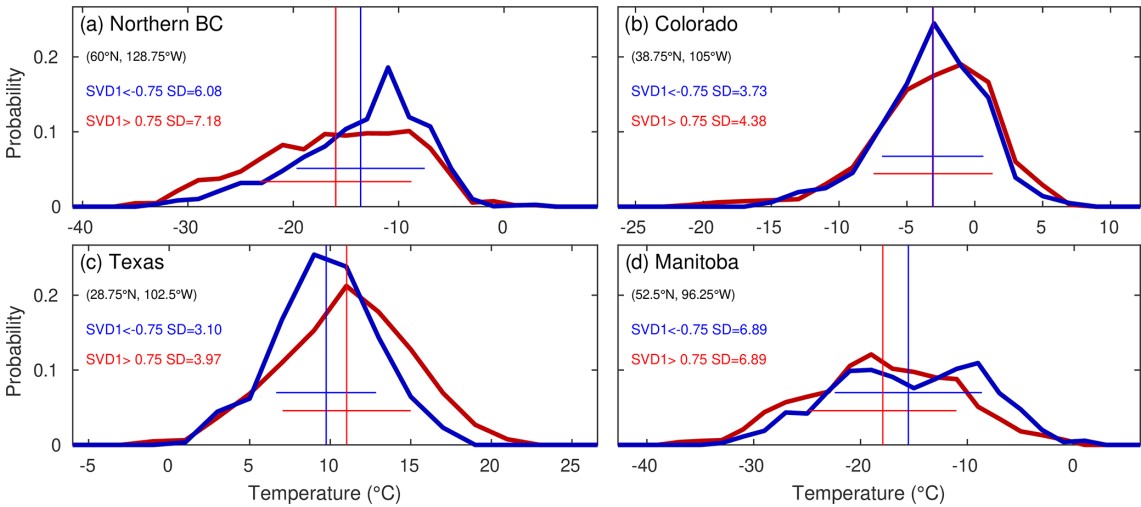

**Figure 11.** Probability density of temperature at four reference locations (indicated with letters in Fig. 10). Statistics are reported separately for SVD1$_{SST} > 0.75$ (red) and SVD1$_{SST} < -0.75$ (blue). The standard deviation (SD) is reported in each panel and illustrated with horizontal lines as intervals of 2 SD. The means of the distributions are indicated with vertical lines.

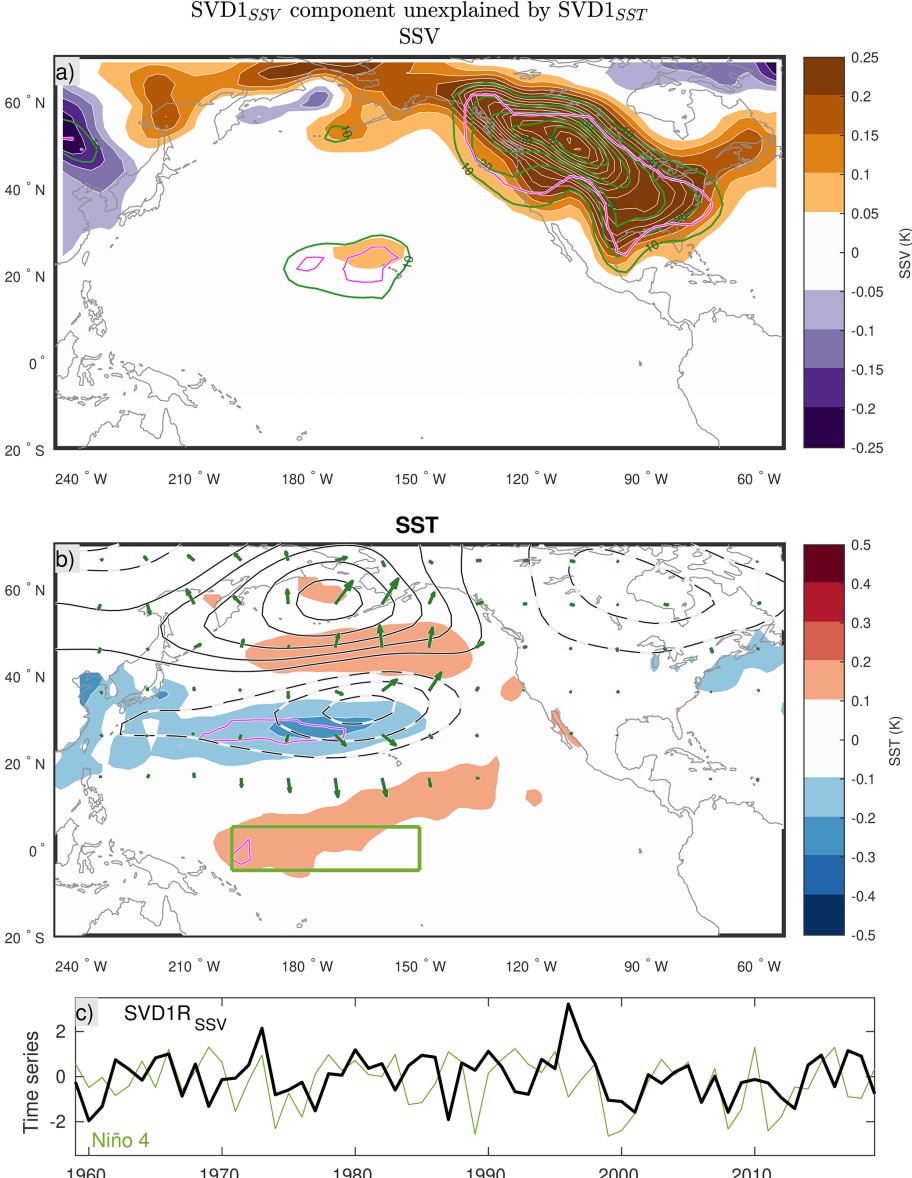

**Figure 12.** Subseasonal (10–60 d) SAT variability (SSV) **(a)** and SST **(b)** are regressed on the component of SVD1$_{SSV}$ that is not correlated with SVD1$_{SST}$ (SVD1R$_{SSV}$). Regions that are statistically significant at the $\alpha_{FDR} = 0.1$ significance level are contoured in magenta. The green contours superimposed on panel **(a)** indicate the fraction of interannual SSV variability explained by this mode with 10 % contour intervals (solid and dashed contours are used for positive and negative values, respectively; the 0 line is omitted). The regression of $Z500$ (contours, 5 m intervals with solid and dashed black contours for positive and negative anomalies) is also shown in panel **(b)**. The corresponding wave-activity fluxes (Takaya and Nakamura, 2001) are shown with green arrows. A distance of 1° corresponds to a flux of $0.125\,\mathrm{m}^2\,\mathrm{s}^{-2}$. The SVD1R$_{SSV}$ time series, as well as the Niño 4 index, are shown in panel **(c)** and the region used to compute this index is illustrated over the SST pattern in panel **(b)**.

and the more out-of-phase relationship of eddy geopotential height and temperature anomalies throughout the depth of the troposphere.TS5

Our analysis thus suggests that, during La Niña winters, the activity of subseasonal eddies is enhanced over the North Pacific as they propagate eastward towards North America. Once they reach the North American coast, these eddies have

strong signatures in lower-tropospheric temperatures due to the reduced near-surface damping of temperature anomalies over land in comparison to over the ocean, thereby enhancing SAT variability. This enhanced SAT variability, combined with cold winter-mean anomalies during La Niña, is associated with an enhanced likelihood of persistent cold extremes, especially over western North America. Our analysis is thus

https://doi.org/10.5194/wcd-3-1-2021 Weather Clim. Dynam., 3, 1–18, 2021

in agreement with the recent finding by Sung et al. (2019) that interdecadal La Niña-like conditions can enhance temperature extremes over North America through modulated baroclinic energy conversion of NPO anomalies, and we have confirmed that similar processes are operative also with interannual ENSO variability.

ENSO-induced anomalies of the extratropical circulation share some similarities with the PNA, but they are not identical (Straus and Shukla, 2002). The anomalies are known, for instance, to be projected also onto the tropical Northern Hemisphere (TNH) pattern (Soulard et al., 2019; Trenberth et al., 1998). It remains unclear at this stage whether the modulation of baroclinic energy conversion is achieved through the projection of the extratropical response on the internally driven PNA or TNH. We nevertheless speculate that it may be achieved primarily through the PNA, since important modulations of baroclinic energy conversion take place over the western North Pacific, where the PNA has a greater influence on the winter-mean flow. A more detailed investigation of the modulations of subseasonal energy sources by internally generated interannual variability should be the topic of a future study. Nonetheless, we briefly investigated other factors that can affect subseasonal variability like ENSO, to find that an important fraction of subseasonal SAT variability over North America appears related to interannual PNA-like or NPO-like atmospheric variability that is uncorrelated with ENSO. Our ability to forecast subseasonal variance over North America may thus depend on the forecast skill of these other atmospheric teleconnections seasons in advance and thus on whether they are externally forced or purely internally generated.

Concerning subseasonal predictions, our results suggest that predictive skill over North America may be deteriorated during La Niña winters due to enhanced energy conversion to subseasonal variability and, as a consequence, increased atmospheric internal variability over the sector. This is in agreement with the overall less skillful predictions achieved during the negative phase of the PNA (Lin and Derome, 1996; Sheng, 2002), which is to some extent similar to the extratropical response to La Niña in terms of extratropical mean flow changes.

*Code availability.* The codes used in this paper can be obtained from the authors upon request.

*Data availability.* JRA-55 (https://doi.org/10.5065/D6HH6H41, Japan Meteorological Agency, 2013 TS6) was obtained from the NCAR/UCAR Research Data Archive (RDA). The HadISST dataset was obtained from the Met Office Hadley Centre (https://doi.org/10.5065/XMYE-AN84, Hadley Centre for Climate Prediction and Research/Met Office/Ministry of Defence/United Kingdom, 2000 TS7).

*Author contributions.* All authors designed the analysis and edited the paper. PM performed the calculations, plotted the results, and drafted the original paper.

*Competing interests.* The authors declare that they have no conflict of interest.

*Acknowledgements.* We are grateful to Hai Lin, Yang Zhang, and an anonymous reviewer for their useful comments on the manuscript.

*Financial support.* This study is supported in part by the Japanese Ministry of Education, Culture, Sports, Science and Technology (MEXT) through the Arctic Challenge for Sustainability (ArCS/ArCS-II) Program (JP-MXD1300000000/JPMXD1420318865) and the Integrated Research Program for Advancing Climate Models (JP-MXD0717935457); by the Japan Science and Technology Agency through Belmont Forum CRA "InterDec" and COI-NEXT (JPMJPF2013); by the Environmental Restoration and Conservation Agency of Japan through the Environment Research and Technology Development Fund (JPMEERF20192004); and by the Japan Society for the Promotion of Science (JSPS) through Grants-in-Aid for Scientific Research JP16H01844, JP18H01278, JP18H01281, JP19H05702, and JP19H05703 (on Innovative Areas 6102). Patrick Martineau is partly supported by Grant-in-Aid for JSPS Research Fellow.

*Review statement.* This paper was edited by Yang Zhang and reviewed by Hai Lin and one anonymous referee.

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

## Remarks from the typesetter

TS1  Please give an explanation of why this needs to be changed. We have to ask the handling editor for approval. Thanks.

TS2  Please confirm.

TS3  Please confirm.

TS4  Please confirm the paragraph formatting.

TS5  Please give an explanation of why this needs to be changed. We have to ask the handling editor for approval. Thanks.

TS6  Please note that the links have to appear in this section as well. Please confirm.

TS7  Please confirm.