# Peer review of "Influence of ENSO on North American subseasonal surface air temperature variability"

_Weather and Climate Dynamics, 2020_

## Short Comment (SC1) · 13 May 2020

I can't add much to the text as I have no doubt that the influence of ENSO on NA temperature variability is strong, and that the qualitative mechanisms that are described apply. But since I have an interest in the underlying fundamental mechanism in the cause of the climate dipoles, there is a different approach that can be applied.

In Chapter 11 of Mathematical Geoenergy, we developed a general formulation based on Laplace's Tidal Equations (LTE) to aid in the analysis of standing wave climate models, focusing on the ENSO and QBO behaviors in the book. As a means of cross-validating this formulation, it makes sense to test the LTE model against other climate indices, such as the northern latitude indices comprised of the Arctic Oscilla-

tion/Northern Annular Mode (AO/NAM) and the Pacific North America (PNA) pattern.

Both the PNA and the Arctic Oscillation can be easily fit from a perturbation of the NAO model, which can be deduced from the known similarity between the AO and NAO — ("The North Atlantic oscillation (NAO) is a close relative of the AO").

Fig 1 shows the common tidal forcing for each of these models, with the LTE modulation in the lower panel. The tidal forcing has a strong semi-annual factor, as does the QBO (see Chapter 11).

The LTE modulation differs subtly between the three, as the multipliers are slightly different for NAO and AO and within ∼15% for PNA. They are in sync at the yellow arrows shown in the lower panel of Fig 1. The LTE modulation is dependent on the fundamental spatial wavenumber defining the dipole, which should be different for each of the regions.

Fig 2 shows the fits for each of the time-series.

You can see how the NAO and AO are vaguely similar and the the PNA is similar but flipped in polarity. It is known that the QBO has a connection to the polar vortex, so the semi-annual commonality between QBO and AO makes some sense.

Figure 1 : Tidal forcing and LTE modulation for the AO and PNA models, alongside the NAO.

Figure 2 : Model fit for NAO, AO and PNA.

———————————————————

[Figure]

**Fig. 1.**

**Fig. 2.**

---

## Referee Comment (RC1) · Anonymous Referee #1 · 7 Jun 2020

In this study, the authors investigated the influence of ENSO on North American sub-seasonal surface air temperature (SAT) variability in boreal winter. The dominant mode of covariability between 10-60 day band-pass filtered SAT variability and winter-mean SST over the North Pacific sector was identified using an SVD analysis. It was found that La Nina conditions tend to enhance the subseasonal SAT variability over western North America. Detailed analysis of energetics of subseasonal edides was carried out. The results are in agreement with previous studies. An interesting finding is that changes in vertical structure of the subseasonal anomalies are important for energy conversion through heat fluxes. The topic is interesting and the manuscript is in general clearly written. It may contribute to the understanding of subseasonal SAT variability and its related extreme events. I think this manuscript is publishable subject to some

minor revisions.

Specific comments: 1. This study deals with ENSO influence on the North American subseasonal SAT. However, the area chosen for the SST in the SVD analysis is mainly the North Pacific region. The resulting seasonal mean SST anomaly has a strong signal in the North Pacific, which may not be related to ENSO. It would be more reasonable to use the tropical Pacific area, e.g., 30S-30N. 2. In the paragraph starting from line 221 and Fig. 7, it should be justified why the two points, Alaska and Colorado, are selected to perform the lagged regression calculation. Is it based on the variance? Are the results sensitive to the choice of base points? Are these subseasonal patterns consistent with previous studies (e.g., Lin 2015)? 3. What is the implication of this study to subseasonal predictions of surface air temperature in North America?

Reference: Lin, H., 2015: Subseasonal variability of North American wintertime surface air temperature. Climate Dyn., 45, 10.1007/s00382-014-2363-6, 1137-1155.

---

## Referee Comment (RC2) · Anonymous Referee #2 · 17 Jun 2020

This study is interested in the influence of ENSO on subseasonal variability in North American SAT during the winter season. Previous studies found that La Nina conditions are associated with enhanced subseasonal SAT variability (SSV) over western North America. Here, the authors use SVD, regression/correlation and composite analyses to investigate how ENSO affects subseasonal variability through modulation of subseasonal eddies - specifically, via changes to the vertical structure of the eddies which have bearing on the amount of baroclinic energy conversion that occurs.

The subject is interesting and relevant to improving our understanding of climate dynamics, as improving near-term climate predictions, and better understanding the large-scale conditions for extreme events. I feel this manuscript could represent a valuable contribution to these areas of research, but would first require substantial revisions

to address a number of scientific and methodological issues that are a bit unclear. First and foremost, while the abstract sounds nicely focused, the rest of the paper seems to mix together a number of different research questions without quite giving the reader enough guidance to connect them (see especially comment #1). The presentation is generally fine, and figures are of good quality, although the captions should include more details so that the reader need not go back to the text to look up information, abbreviations, etc.

MAJOR COMMENTS

1. The setup of a clear, motivating question in the introduction is not quite there. There seem to be several trains of thought, including the influence of ENSO on SSV, the link from SSV to extremes, and the mechanisms by which ENSO affects SSV, but they are not well connected and in some cases, we seem to be missing some background information needed to make this connection. Some specific comments:

- Is the ENSO-related SSV signal just part of the PNA-related SSV? if not, how is it different?

- How important is the portion of extratropical SSV related to ENSO? It seems key to establish this up-front, since later on in Fig. 10, you show an SSV signal unrelated to your ENSO index (SVD1) that is both substantial in amplitude and very similar to the ENSO-related signal. It would also be nice to show the SSV climatology for reference, perhaps early on in the results section, since this is a field many readers will not be so familiar with.

- And perhaps even one step before this, how important is SAT variability in the 10-60 day band?

- The paragraph starting on L42 seems to be off-topic - if this is meant to relate to the issue of extremes, the connection needs to be made better. In general, the parts of the manuscript dealing with extremes seems like somewhat of an afterthought - it probably

should be either expanded or de-emphasized. Also, the topic sentence seems to say there is a clear association between ENSO and blocking, while later in the paragraph, we see that the association is not clear.

- The paragraph starting on L60 - I'm having some trouble with the logical flow in the first few sentences.

- I'm not sure how familiar most readers are with the term "subseasonal eddies".

2. The title and abstract talk about ENSO's influence on SSV, but the "first step" (L73) is identifying the dominant mode of covariability between North Pacific SST and SSV. Why not use an ENSO index - either one of the standard ones in Table 1, or an EOF-based index of tropical Pacific SST (Takahashi et al., Cai et al.)? I see that the SVD1 produces indices that are well correlated with ENSO, but I don't understand the point of using this over using actual ENSO indices (perhaps there is a good reason but I've missed it, in which case it should be better explained). Even if one were to use an SVD, would it not be better to choose a tropical Pacific box for the SST field? It's been shown that including the North Pacific mixes frequencies and forcing source regions (Wills et al.).

- Cai et al. (2018): Increased variability of eastern Pacific El Niño under greenhouse warming. Nature, 564

- Takahashi et al. (2011): ENSO regimes: Reinterpreting the canonical and Modoki El Niño. Geophys. Res. Lett., 38, L10704

- https://agupubs.onlinelibrary.wiley.com/doi/full/10.1002/2017GL076327

3. I like the dynamical line of investigation regarding why subseasonal eddies may be more "active" during La Nina. I think the argumentation could be made more convincing, and this would really strengthen the paper as a whole. First, the connection from subseasonal SAT variability to the subseasonal eddies should be made clearer in the text (just a few lines of explanation to help the reader interpret the figures). Second,

Fig. 8 is not so compelling as a demonstration that differences in the vertical structure of the eddies are key. Some suggestions: (a) show a larger longitudinal range that include all the positive/negative centres of action seen in Fig. 7, so we see the change in vertical structure systematically with each one, (b) show barotropic energy conversion with height, so we see the big increases where the temperature and Z fields are most offset.

4. Some of the analysis choices seem rather arbitrary, and need to be better explained. Also the analysis itself. Some examples, but not exhaustive:

- regions for the SVD (the SSV box is probably related to target area and climatological field, but what about SST? see comment #2)

- locations for temperature regressions in Fig. 7

- location for Z regressions in Fig. 8. Presumably, we want to look at the eddies responsible for SAT variability such as that seen in Fig. 7? It would be helpful to justify this point and mark it on one of the maps.

- what frequency data is used for the various analyses? Presumably daily or 6-hourly for SSV that is then band-pass filtered? The SVD looks to be using monthly or seasonal averages? What about in the regressions for vertical structures?

- how are warm/cold extremes identified?

- how are u", v", T" defined?

- Fig. 7: is the SAT index using the 10-60 day filtered field? Is Z filtered?

5. The composites need some measure of significance, either via comparison to the total variability, or via comparison to the inter-composite spread, or via some boostrapping, etc. This is especially important in light of the fact that internal variability seems to play an important role in shaping extratropical ENSO teleconnections. Also, how many "samples" (days, months, seasons?) make up each composite? https://journals.ametsoc.org/jcli/article/31/13/4991/92604/How-Well-Do-We-Know-ENSO-s-Climate-Impacts-over

OTHER POINTS

- Fig. 1: show SSV climatology?

- Fig. 2, 3, 4: please define abbreviations in caption

- L257: "an important asymmetry" - isn't this just a consequence of L252-254?

- L262: "which significantly widens the probability distribution..." I don't understand this explanation. Are you suggesting that a wider distribution mean that you "lose" extremes on one end but not the other, and if so, why?

- L286: EN and LN flipped?

---

## Author Comment (AC1) · 23 Sep 2020

**Response to reviewers**

We would like to thank both reviewers for taking the time to review our manuscript and provide valuable comments. These comments will, without any doubt, help us improve our analysis and discussion. In the following, we describe how we intend to revise our manuscript to address these comments. This response will be followed by a more detailed point-by-point response to all the reviewer's comments.

**1) Choice of the SST domain in the SVD analysis**

Both reviewers have noted that the current SST domain used for the SVD analysis, which extends over most of the equatorial and northern sectors of the Pacific Ocean, is likely too broad to adequately capture ENSO's relationship with North American subseasonal variability. One issue is the possible contamination by other timescales of variability affecting the North-Pacific sector. In response to these comments, we will limit the SST domain to 20°S - 20°N. Our results are, however, found rather insensitive to this change of domain, and thus our conclusions on the earlier version all remain valid.

**2) Purpose of using the SVD analysis**

Reviewer #2 also expressed doubts about the practicality of using the SVD analysis over the use of classical ENSO indices. We believe however that the SVD analysis is useful for objectively assessing which flavor of ENSO is optimally related to subseasonal variability over North America through the maximization of the covariance. Starting from the classical ENSO indices would require repeating the analysis for each index, and it does not necessarily guarantee that these indices could uncover a parcicular SST pattern having the optimal connection with the subseasonal variability. Identifying this optimal connection via the SVD anlaysis thus also contributes to improving the significance of our subsequent analyses by providing time series that are the most representative of the relationship between equatorial SST variability and subseasonal variability over North America.

**3) The implication for North American weather predictability**

Reviewer # 1 suggested discussing the implications of our results for weather predictability over North America. Since we agree this would add value to the manuscript, we will add the following discussion to our conclusion section to the revised manuscript: "Our results suggest that predictive skill over North America may be deteriorated during La Niña due to enhanced baroclinic energy conversion to SSV and, as a consequence, increased atmospheric internal variability over the sector. This is in agreement with the overall less skillful predictions achieved during the negative phase of the PNA (Lin and Derome 1996; Sheng 2002) which, in terms of extratropical mean flow changes, is to some extent similar to the extratropical response to La Niña."

**4) The sensitivity of wave train structures to reference locations**

In response to a comment from reviewer # 1, it will be added to the revised manuscript that the locations are chosen because the identified mode of covariability has a large impact on SSV over these sectors (Fig. 2) and atmospheric circulation patterns associated with localized SAT variability are quite sensitive to a parcicular choice of the reference locations. It will also be noted in the revised manuscript that these patterns share similar features, such as their spatial scale and meandering, with the circulation anomalies associated with the leading modes of SAT variability (Lin 2015) although the exact

location of their cyclonic and anticyclonic centres of action are not the same. We consider that they may correspond to modes of lesser importance or combinations of the leading modes.

**5) Relationship with the PNA**

Reviewer # 2 inquired whether the extratropical response is the one associated with the well-known PNA pattern. The observed winter-mean response is in fact more similar to the extratropical response forced by ENSO than the one associated with the purely internally-generated PNA (Straus and Shukla 2002). It is also known, for instance, to project onto the Tropical Northern Hemisphere (TNH) pattern (Trenberth et al. 1998; Soulard et al. 2019). It remains unclear whether the modulation of baroclinic energy conversion is achieved through the projection of the extratropical response on the internally-driven PNA or TNH, but we speculate that it may be achieved through the PNA since important modulations of baroclinic energy conversion take place over the western North Pacific where the PNA has a greater influence on the mean flow. We believe this is an important point, and the relevant discussions will be added to the revised manuscript.

It is, however, out of the scope of the present study to assess quantitatively whether this impact is achieved through a projection on natural modes of variability. It should be the topic of our future standalone paper on internally-generated variability, in which the PNA, the TNH, and other teleconnections affecting the North Pacific sector should be investigated.

**6) Importance of ENSO's influence on SSV**

Reviewer # 2 has proposed to discuss the importance of ENSO's influence on SSV. To this end, we will add a new figure (Fig.1) to show the climatology of SSV and its interannual modulations. We will also illustrate the fraction of this variability modulated by ENSO on our Fig. 2 showing the result of the SVD analysis.

[Figure]

New Figure 1: *Climatological features of SAT variability (1958-2015). The a) DJF-mean SAT, b) DJF-mean subseasonal SAT variability, or SSV, c) DJF-mean high-frequency SAT variability, and d) interannual modulation of subseasonal SAT variability are shown with colour shadings. All variables are in units of K. Variability is illustrated with the standard deviation.*

**7) Improving the focus of the introduction**

Reviewer # 2 noted that one of our introductory paragraphs, where we discuss blocking events and weather extremes, seems off-topic. The reason we discuss blocking there is to illustrate, with a specific event of subseasonal time scale, how ENSO can modulate subseasonal variability and associated weather impacts. Such influence has rarely been discussed for general subseasonal variability, but abundant literature investigated this influence in the context of atmospheric blockings. We believe that our discussion with linkage to extreme events in referring to blocking is relevant to subseasonal variability, because extremes are, by definition, associated with highly anomalous, and thus variable, weather. We nevertheless agree that this discussion could be connected better with the preceding part of the introduction, and we will modify the text to improve this connection.

Also, we believe that the section about the impact of ENSO on extremes, although brief, may be of interest to some readers. We consider it important to illustrate that subseasonal variability is not the only factor controlling the occurrence of extremes. The winter-mean changes in temperature associated with ENSO play a very important role. Nonetheless, we will de-emphasize this aspect by removing the last paragraph of the introduction that summarizes this finding.

**8) Assessing modulations of the vertical structure of subseasonal eddies**

Reviewer # 2 expressed doubts about the utility of our analysis of the vertical structure of eddies and thus suggested looking at all cyclonic/anticyclonic anomaly centers associated with subseasonal eddies instead of focusing on the circulation near the reference grid points. The reason we keep our focus close to the reference grid points is that it is where we have greater confidence about the structure of the subseasonal eddies. Farther away from the reference grid points, where remote anticyclonic and cyclonic anomalies could be located, correlation and regression values decrease substantially, which means we do not have as much confidence in the identified structure under the low signal to noise ratio. This is why we have chosen a reference grid point located in a region of large modulations of CP, but not over North America, for this analysis. We will revise the manuscript to better justify this approach.

We greatly appreciate the reviewer's suggestion to illustrate how structural changes of subseasonal eddies affect baroclinic energy conversion. To this end, we will add panels showing the heat fluxes associated with subseasonal eddies to illustrate how the overall contributions to poleward heat fluxes tend to increase in $SVD1_{SST} > 1$, especially to the west of the reference longitude due to the enhanced baroclinic structure of the eddies.

[Figure]

Figure 9: (a, b) Zonal sections of subseasonal anomalies in geopotential height (black contours with intervals of 10 m; solid and dashed lines are used for positive and negative values, respectively) and temperature (shadings) both regressed onto the reference time series of geopotential height at [53°N, 165.5°W] (green circles on Figs. 3c and 6), for winters when (a; left) SVD1$_{SST}$ > 1 and (b, right) SVD1$_{SST}$ < − 1. The maxima of the height and temperature anomalies at individual pressure levels are connected verticvally with green and magenta lines, respectively. The lines shown in b) are also repeated in grey in a) for comparison. (c, d) The associated meridional heat fluxes ($v'T'$) are contoured at intervals of 1 m K s$^{-1}$ with red and blue lines for positive (northward) and negative (southward) values, respectively.

**9) Statistical significance**

We will follow Reviewer # 2's suggestion to add statistical significance to Figs. 5 and 6. Statistical significance will be assessed through a bootstrapping approach with randomly resampled (1500 times) composites of the same sample sizes as those shown in Figs. 5 and 6.

---

## Author Comment (AC2) · 23 Sep 2020

Dear Dr. Pukite, Thank you very much for your comments. Could the theory described in your book be used to study the modulation of quasi-stationary Rossby waves by ENSO, the PNA, and the NAO?
* * *

---

## Author Response (AR1)

**Response to reviewers: Influence of ENSO on North American subseasonal surface air temperature variability (wcd-2020-22)**

We are grateful to the two reviewers for providing valuable comments on our manuscript. Our responses are included below.

**Response to reviewer 1**

We thank the reviewer for taking the time to assess our manuscript and for providing valuable comments. As suggested by the two reviewers, we modify our analysis to focus on tropical Pacific SST variability instead of the whole NH Pacific. Our results are not sensitive to the change of domain and thus our conclusions remain the same. Our replies to the reviewer's comments are shown in blue below. Line numbers refer to the manuscript with tracked changes.

In this study, the authors investigated the influence of ENSO on North American subseasonal surface air temperature (SAT) variability in boreal winter. The dominant mode of covariability between 10-60 day band-pass filtered SAT variability and winter-mean SST over the North Pacific sector was identified using an SVD analysis. It was found that La Nina conditions tend to enhance the subseasonal SAT variability over western North America. Detailed analysis of energetics of subseasonal edides was carried out. The results are in agreement with previous studies. An interesting finding is that changes in vertical structure of the subseasonal anomalies are important for energy conversion through heat fluxes. The topic is interesting and the manuscript is in general clearly written. It may contribute to the understanding of subseasonal SAT variability and its related extreme events. I think this manuscript is publishable subject to some minor revisions.

Specific comments: 1. This study deals with ENSO influence on the North American subseasonal SAT. However, the area chosen for the SST in the SVD analysis is mainly the North Pacific region. The resulting seasonal mean SST anomaly has a strong signal in the North Pacific, which may not be related to ENSO. It would be more reasonable to use the tropical Pacific area, e.g., 30S-30N.

Thank you very much for the comment. As suggested, we modify the sector over which SST is used for the SVD analysis. By limiting the domain to 20S-20N, we now focus solely on tropical Pacific variability. Please note that extratropical SST anomalies are still present, albeit weaker than the equatorial anomalies (Fig. 2). This is not surprising since tropical SST variability related to ENSO is accompanied by SST variability in the extratropics (e.g., Deser et al. 2010).

2. In the paragraph starting from line 221 and Fig. 7, it should be justified why the two points, Alaska and Colorado, are selected to perform the lagged regression calculation. Is it based on the variance? Are the results sensitive to the choice of base points? Are these subseasonal patterns consistent with previous studies (e.g., Lin 2015)?

We add at line 293: "These locations are chosen because SVD1 has a large impact on SSV over these sectors (Fig. 2)". We also add at line 304: "As illustrated in Fig. 8, atmospheric circulation patterns associated with localized SAT variability are quite sensitive to the reference location. These patterns share similar features, such as their spatial scale and meandering, with the circulation anomalies associated with the leading modes of SAT variability (Lin 2015). The exact location of their cyclonic and anticyclonic centres of action are however not the same. They may correspond to modes of lesser importance or combinations of the leading modes. Other reference locations over North America were assessed and revealed different circulation anomalies (not shown)."

**3. What is the implication of this study to subseasonal predictions of surface air temperature in North America?**

We add the following discussion at the end of our manuscript: "Concerning subseasonal predictions, our results suggest that predictive skill over North America may be deteriorated during La Niña winters due to enhanced energy conversion to subseasonal variability and, as a consequence, increased atmospheric internal variability over the sector. This is in agreement with the overall less skillful predictions achieved during the negative phase of the PNA (Lin and Derome 1996; Sheng 2002) which, in terms of extratropical mean flow changes, is to some extent similar to the extratropical response to La Niña."

Reference: Lin, H., 2015: Subseasonal variability of North American wintertime surface air temperature. Climate Dyn., 45, 10.1007/s00382-014-2363-6, 1137-1155.

**References**

- Deser, C., M. A. Alexander, S.-P. Xie, and A. S. Phillips, 2010: *Sea Surface Temperature Variability: Patterns and Mechanisms*. 115–143 pp.
- Lin, H., 2015: Subseasonal variability of North American wintertime surface air temperature. *Clim. Dyn.*, **45**, 1137–1155, https://doi.org/10.1007/s00382-014-2363-6.
- ——, and J. Derome, 1996: Changes in predictability associated with the PNA pattern. *Tellus A Dyn. Meteorol. Oceanogr.*, **48**, 553–571, https://doi.org/10.3402/tellusa.v48i4.12139.
- Sheng, J., 2002: GCM experiments on changes in atmospheric predictability associated with the PNA pattern and tropical SST anomalies. *Tellus A Dyn. Meteorol. Oceanogr.*, **54**, 317–329, https://doi.org/10.3402/tellusa.v54i4.12153.

**Response to reviewer 2**

We thank the reviewer for providing useful comments which have helped us improve our manuscript. As suggested by the two reviewers, we modify our analysis to focus on tropical Pacific SST variability. Our results are not sensitive to the change of domain and thus our conclusions remain the same. Our responses to the comments are written in blue below. Figure numbers refer to the new figure set which includes a new figure at the beginning.

This study is interested in the influence of ENSO on subseasonal variability in North American SAT during the winter season. Previous studies found that La Nina conditions are associated with enhanced subseasonal SAT variability (SSV) over western North America. Here, the authors use SVD, regression/correlation and composite analyses to investigate how ENSO affects subseasonal variability through modulation of subseasonal eddies - specifically, via changes to the vertical structure of the eddies which have bearing on the amount of baroclinic energy conversion that occurs. The subject is interesting and relevant to improving our understanding of climate dynamics, as improving near-term climate predictions, and better understanding the large-scale conditions for extreme events. I feel this manuscript could represent a valuable contribution to these areas of research, but would first require substantial revisions to address a number of scientific and methodological issues that are a bit unclear.

First and foremost, while the abstract sounds nicely focused, the rest of the paper seems to mix together a number of different research questions without quite giving the reader enough guidance to connect them (see especially comment #1). The presentation is generally fine, and figures are of good quality, although the captions should include more details so that the reader need not go back to the text to look up information, abbreviations, etc.

**MAJOR COMMENTS**

The setup of a clear, motivating question in the introduction is not quite there. There seem to be several trains of thought, including the influence of ENSO on SSV, the link from SSV to extremes, and the mechanisms by which ENSO affects SSV, but they are not well connected and in some cases, we seem to be missing some background information needed to make this connection.

**Some specific comments:**

**- Is the ENSO-related SSV signal just part of the PNA-related SSV? if not, how is it different?**

This is a good point. We now note at line 221 that the winter-mean response is "more similar to the extratropical response forced by ENSO than the internally-generated PNA (Straus and Shukla 2002)" and we add the following discussion in the conclusion (line 424) :" ENSO-induced anomalies of the extratropical circulation share some similarities with the PNA, but they are not identical (Straus and Shukla, 2002). The anomalies are known, for instance, to be projected also onto the Tropical Northern Hemisphere (TNH) pattern (Soulard et al., 2019; Trenberth et al., 1998). It remains unclear at this stage whether the modulation of baroclinic energy conversion is achieved through the projection of the extratropical response on the internally-driven PNA or TNH. We nevertheless speculate that it may be achieved primarily through the PNA, since important modulations of baroclinic energy conversion take place over the western North Pacific, where the PNA has a greater influence on the winter-mean flow. A more detailed investigation of the modulations of subseasonal energy sources by internally-generated internanual variability should be the topic of a future study".

In summary, we believe that while it is interesting to understand how the teleconnections onto which ENSO's extratropical response is projected modulate subseasonal variability, it is out of the scope of the

present study and should be the topic of a standalone paper on internally-generated variability. Not only the PNA but also the TNH and other teleconnections affecting the North Pacific sector should be investigated.

- How important is the portion of extratropical SSV related to ENSO? It seems key to establish this upfront, since later on in Fig. 10, you show an SSV signal unrelated to your ENSO index (SVD1) that is both substantial in amplitude and very similar to the ENSO-related signal. It would also be nice to show the SSV climatology for reference, perhaps early on in the results section, since this is a field many readers will not be so familiar with.

Thank you for the comment. Instead of showing the percentage of the climatology that these ENSOrelated variations represent, we show the percentage of local total interannual variance explained by ENSO (Fig. 2a). The percentage explained is up to about 10% which contrasts with the variance explained by the ENSO-unrelated signal which explains up to 50% of the total variance (Fig. 11a). These percentages of variance explained are now indicated in the manuscript.

Also, we add a new figure (Fig. 1) to show the basic climatological properties of SAT. The climatologies of SAT and SSV are now discussed at lines 184-203.

- And perhaps even one step before this, how important is SAT variability in the 10-60 day band?

We add a discussion (lines 192-199) on the importance of SAT variability in the 10-60 day band by contrasting variability at this time scale with high-frequency variability (2-8 days) which is more thoroughly studied. To this end, the climatology of high-frequency SAT variability is shown in Fig. 1c.

- The paragraph starting on L42 seems to be off-topic - if this is meant to relate to the issue of extremes, the connection needs to be made better. In general, the parts of the manuscript dealing with extremes seems like somewhat of an afterthought - it probably should be either expanded or de-emphasized.

The reason we discuss blocking there is to illustrate, with a specific event of subseasonal time scale, how ENSO can modulate subseasonal variability and associated weather impacts. Such influence has rarely been discussed for general subseasonal variability, but abundant literature investigated this influence in the context of atmospheric blockings. We believe that our discussion with linkage to extreme events in referring to blocking is relevant to subseasonal variability, because extremes are, by definition, associated with highly anomalous, and thus variable, weather. We nevertheless agree that this discussion could be connected better with the preceding part of the introduction. We try to improve the link with the preceding paragraph by writing (line 45): "This ENSO influence on intraseasonal variability may be achieved in part through modulation of the frequency of blocking events..." Afterward, we introduce in more detail their relationship with weather extremes.

We believe that the section about the impact of ENSO on extremes, although brief, may be of interest to some readers. We consider that it is important to illustrate that subseasonal variability is not the only factor controlling extremes. Its combined influence with ENSO-related winter-mean changes in temperature is important. Nonetheless, we de-emphasize this aspect by removing the last paragraph of the introduction that summarizes this finding.

Also, the topic sentence seems to say there is a clear association between ENSO and blocking, while later in the paragraph, we see that the association is not clear.

We soften the topic sentence and discuss more extensively a possible reason for the different conclusions among studies (line 56): "The studies that have defined blocking events as prominent anomalies have reported an increase in the frequency of blocking during La Niña (Renwick and Wallace 1996; Barriopedro and Calvo 2014; Chen and van den Dool 1997), which is in agreement with the aforementioned changes in intraseasonal variability. Considering the link between blocking and weather extremes, this suggests a potential increase in the frequency of extreme cold episodes on subseasonal time scales during La Niña."

- The paragraph starting on L60 - I'm having some trouble with the logical flow in the first few sentences.

We rewrite the first few sentences to improve clarity (lines 71-75).

- I'm not sure how familiar most readers are with the term "subseasonal eddies".

We now define what we mean by subseasonal eddies at line 96: "atmospheric circulation anomalies on subseasonal time scales, hereafter referred to as subseasonal eddies".

2. The title and abstract talk about ENSO's influence on SSV, but the "first step" (L73) is identifying the dominant mode of covariability between North Pacific SST and SSV. Why not use an ENSO index - either one of the standard ones in Table 1, or an EOFbased index of tropical Pacific SST (Takahashi et al., Cai et al.)? I see that the SVD1 produces indices that are well correlated with ENSO, but I don't understand the point of using this over using actual ENSO indices (perhaps there is a good reason but I've missed it, in which case it should be better explained). Even if one were to use an SVD, would it not be better to choose a tropical Pacific box for the SST field? It's been shown that including the North Pacific mixes frequencies and forcing source regions (Wills et al.).

- Cai et al. (2018): Increased variability of eastern Pacific El Niño under greenhouse warming. Nature, 564

- Takahashi et al. (2011): ENSO regimes: Reinterpreting the canonical and Modoki El Niño. Geophys. Res. Lett., 38, L10704

- https://agupubs.onlinelibrary.wiley.com/doi/full/10.1002/2017GL076327

Thank you very much for the comment. We focus our new analysis on tropical SST variability by limiting the SVD domain to 20°S-20°N. Our results are not sensitive to this modification and our general conclusions remain the same.

We now better motivate our approach in the Methods section (lines 111-119): "One approach to investigating the influence of ENSO on subseasonal SAT variability is to start from classic ENSO indices (see the next section). The individual indices, however, represent different "flavours" of ENSO that may exert distinct impacts on North-American subseasonal SAT variability. Instead of repeating our analysis for all these indices, we identify, via singular value decomposition (SVD) analysis (Bjornsson and Venegas, 1997; Bretherton et al., 1992), a particular "flavour" of tropical Pacific SST variability that is optimally related to subseasonal SAT variability over North America. Identifying this optimal influence is not only important to better predict SAT variability from SST anomalies but also contributes to improving the clarity of the rest of our analyses by focusing on the strongest statistical connection."

3. I like the dynamical line of investigation regarding why subseasonal eddies may be more "active" during La Nina. I think the argumentation could be made more convincing, and this would really

strengthen the paper as a whole. First, the connection from subseasonal SAT variability to the subseasonal eddies should be made clearer in the text (just a few lines of explanation to help the reader interpret the figures).

We add a discussion to justify why we study energetics to better understand SAT variability at lines 230-232): "The rationale is that SSV is produced by weather systems (or eddies) that have deep structures within the troposphere and thus better understanding of interannual fluctuations of SSV can be acquired through investigating year-to-year changes in processes that energize these eddies".

Second, Fig. 8 is not so compelling as a demonstration that differences in the vertical structure of the eddies are key. Some suggestions: (a) show a larger longitudinal range that include all the positive/negative centres of action seen in Fig. 7, so we see the change in vertical structure systematically with each one, (b) show barotropic energy conversion with height, so we see the big increases where the temperature and Z fields are most offset.

We keep our focus close to the reference latitude because it is where we have greater confidence about the structure of the subseasonal eddies. Further away from the reference time series, where remote anticyclonic and cyclonic anomalies are located, correlations and regressions decrease substantially which means we do not have as much confidence in the identified structure since the signal to noise ratio is low. This is why we have chosen a reference time series located in the region of large modulation of CP, and not over North America, for this analysis. We add (line 320): "For a robust illustration of the structure of eddies over that sector, it is preferable to use a local reference grid point in this analysis (as indicated with a green circle in Figs. 3c and 6). Eddy structures constructed from remote reference SAT time series over Alaska and Colorado (Fig. 8) do exhibit signals over the North Pacific, but the correlations are substantially weaker which indicates a low signal to noise ratio." In Fig. 8, we add the heat fluxes associated with subseasonal eddy structures to illustrate how the overall contributions to poleward heat fluxes are enhanced in  $SVD1_{SST} > 1$  and add the following discussion (line 330): "The net meridional heat fluxes associated with these structures (Figs. 9c-d) are obviously larger for  $SVD1_{SST} > 1$ due to marked enhancement (approximate doubling) of poleward heat transport to the west of the reference longitude, which cannot be offset completely by a slight increase in the southward transport to the east

4. Some of the analysis choices seem rather arbitrary, and need to be better explained. Also the analysis itself. Some examples, but not exhaustive:

- regions for the SVD (the SSV box is probably related to target area and climatological field, but what about SST? see comment #2)

In our revised manuscript we limit the SST domain used for the SVD analysis to 20°S-20°N following the reviewers' comments. This is motivated by our goal to investigate the influence of tropical Pacific variability on North American SSV.

- locations for temperature regressions in Fig. 7

The reference locations used for the regressions shown in Fig. 8 are now indicated with magenta Xs.

- location for Z regressions in Fig. 8. Presumably, we want to look at the eddies responsible for SAT variability such as that seen in Fig. 7? It would be helpful to justify this point and mark it on one of the maps.

The reference location used for the regressions shown in Fig. 9 is now indicated with green circles in Figs. 3 and 6.

- what frequency data is used for the various analyses? Presumably daily or 6-hourly for SSV that is then band-pass filtered? The SVD looks to be using monthly or seasonal averages? What about in the regressions for vertical structures?

We now indicate that we use 6-hourly data for SSW and that the standard deviation is taken over DJF. The section describing the SVD analysis (2.2) states that winter-mean SST is used. We add that the regressions for vertical eddy structures are based on 10-60 day bandpass filtered data.

**- how are warm/cold extremes identified?**

The definition of cold/warm extremes is described in section 3.5: "Cold (warm) extreme days are defined as the days when 10-day lowpass-filtered SAT anomaly falls below (rises above) the 5th (95th) percentiles at each grid point over the 58 winters. Their frequency, calculated as the percentage of winter days each year, is then regressed onto the SVD1SST time series".

**- how are u", v", T" defined?**

We add in section 2.4 that a 10-day high-pass filter is used.

**- Fig. 7: is the SAT index using the 10-60 day filtered field? Is Z filtered?**

**We add that all the time series are filtered with a 10-60 day bandpass filter before the regressions.**

5. The composites need some measure of significance, either via comparison to the total variability, or via comparison to the inter-composite spread, or via some boostrapping, etc. This is especially important in light of the fact that internal variability seems to play an important role in shaping extratropical ENSO teleconnections. Also, how many "samples" (days, months, seasons?) make up each composite? https://journals.ametsoc.org/jcli/article/31/13/4991/92604/How-Well-Do-WeKnow-ENSO-s-Climate-Impacts-over

We add statistical tests based on bootstrapping to Figs. 5 and 6 and explain in the text (line 255) that "Statistical significance is assessed through a bootstrapping approach with randomly resampled (1500 times) composites of the same sample size as those shown in Fig. 5"

We add that 21 and 14 winters are used for  $SVD1_{SST} > 0.5$  and  $SVD1_{SST} < -0.5$ , respectively. These winters are indicated with Xs and 0s in Fig. 2.

**OTHER POINTS**

**- Fig. 1: show SSV climatology?**

We add a new figure (Fig. 1) to show some basic climatological features of SAT and its subseasonal variability. These features are now discussed at the beginning of the results section (lines 184-203).

**- Fig. 2, 3, 4: please define abbreviations in caption**

We add definitions of the abbreviations of energetics terms in each of these captions.

**- L257: "an important asymmetry" - isn't this just a consequence of L252-254?**

We notice this terminology can be confusing. To clarify we add (line 346): "However, an important mismatch is observed in the response of warm and cold extremes over western and southern North America (Figs. 10a-b). In these sectors, increases in the frequency of cold extremes are not matched with similar decreases in the frequency of warm extremes and *vice-versa*."

- L262: "which significantly widens the probability distribution..." I don't understand this explanation. Are you suggesting that a wider distribution mean that you "lose" extremes on one end but not the other, and if so, why?

To clarify we add the following discussion (lines 354-357): "For instance, if the winter-mean temperature is warmer, one may expect the whole probability distribution of temperature to shift towards warmer temperatures and thereby increase the likelihood of warm extremes and decrease the likelihood of cold extremes. However, if subseasonal variability is enhanced, it can contribute to increasing the frequency of cold extremes, opposing the effect of the changes in the mean"

**- L286: EN and LN flipped?**

Thank you very much for noticing the mistake.

**2.4 Energetics of subseasonal eddies**

Atmospheric energetics (Lorenz, 1955; Oort, 1964) are used to assess how ENSO modulates the sources of energy sustaining

- 140 circulation anomalies that to produce subseasonal SAT variability (or SSV). Energies and their conversion/generation terms are integrated vertically from the surface to 100 hPa for subseasonal variability that has been extracted by applying a 10-60 day bandpass filter to the 6-hourly data (denoted with primes in the following equations). The basic state, denoted with overbars, is defined as the winter-mean (DJF) fields for individual years, which that include seasonal-mean fluctuations related to ENSO variability.
- 145 The eddy available potential energy (EAPE), is defined as

$$EAPE = \gamma^{-1} \frac{T^{\prime 2}}{2}, \tag{1}$$

where  $\gamma$  is a stability parameter defined as

$$\gamma = \frac{p}{R} \left( \frac{R\hat{T}}{C_p p} - \frac{\partial \hat{T}}{\partial p} \right). \tag{2}$$

Here *R* is the gas constant for dry air (287 J K-1 kg-1) and  $C_p$  is the specific heat at constant pressure (1004 J K-1 kg-1). The stability parameter is here based on temperature averaged over the Northern Hemisphere (denoted by the hat operator). The EAPE is proportional to temperature variance when averaged over a season and receives a strong contribution from the lower troposphere where subseasonal temperature anomalies are strongest (not shown).

Several sources of EAPE are considered. The first is baroclinic energy conversion (CP):

$$CP = -\gamma^{-1} \left( u'T' \frac{\partial T}{\partial x} + v'T' \frac{\partial T}{\partial y} \right).$$
(3)

[revised manuscript text omitted]
}$ | 0.09    | 0.13  | 0.14    | 0.23  | SVD1R SSV | 0.00    | 0.03  | 0.04    | 0.15  |
|     |               |         |       |         |       |                      |         |       |         |       |

---

## Author Response (AR2)

We thank the co-editor and Referee #2 for their useful comments in this second round of revision. To address these comments, some substantial changes are made to the analysis. One of them is the extension of the analysis to 2019, which contributes to increasing the robustness of the results. Another is to use field significance whenever statistical significance is tested. We also standardize the threshold for separating SVD1$_{SST}$+ years from SVD1$_{SST}$- years to a standard deviation of ±0.75. Note that we changed the order of Figs. 8 and 9 to improve the flow of the manuscript.

While working on this revision, we have noticed some unfortunate mistakes. A minor one is that wave activity fluxes shown in Fig. 7 are for 300 hPa instead of 500 hPa. A more major one is that in order to separate the years when SVD1 is positive from those when it is negative in the analyses shown in Figs. 8 and 9 we used SVD1$_{SSV}$ instead of SVD1$_{SST}$ as stated in the text. After correcting this mistake, and adjusting the reference location to where changes in correlation between v' and T' are most significant according to field significance (Fig. 6), differences in eddy structures became less important. Nonetheless, we choose to keep this analysis to illustrate that subtle changes in eddy structures can lead to the observed changes in energy conversion.

**Co-Editor Comments:**

1. In section 3.2, the authors need to make a clearer statement on the relation between ENSO and SVD1_SST, which also refers to referee #2's major comment 1. Based on the results in Tab1, correlations between the time series of SVD1_SST and Nino 3 (Nino 3.4) index already reach -0.96 (-0.94), suggesting that SVD1_SST mostly reflects the variability of ENSO. I think this is the main rationale that SVD1_SSV (and the authors' following analysis) can greatly represents ENSO's influence. However, compared to this evident result, the authors' argument on the relation between ENSO and SVD1_SST in the text seems a bit weak and less straightforward.

We now make a stronger statement by adding the following (in bold): Indeed, the time series representing the temporal variability of this pattern (SVD1SST, Fig. 2c) is strongly anticorrelated to all the four Niño indices (Table 1), which indicates that SVD1$_{SST}$ essentially reflects SST variability associated with ENSO.

2. In section 3.5, I suggest the authors revise their argument on the relation between winter-mean state and extreme temperature events (i.e. line 321-322), which also refers to referee #2's major comment 4. The winter-mean state is the average of the day-by-day temperature in winter season, in which the extreme events always make considerable contribution. Thus, the mean state and extreme events are intrinsically coupled. Merely from Figure 10, it is not convincing enough to conclude that the mean state sets the frequency of the extremes.

We agree with the co-editor and referee #2. We cannot causally link the winter-mean and the frequency of extremes, because the winter mean is itself an average of all days, including the extremes. We thus avoid implying causality and rather aim to discuss changes in the properties of the probability distribution of surface temperature associated with the observed changes in the winter-mean, variability, and frequency of extremes.

3. Line 161-162: the domain integrated energy flux term might be small over some region but not zero. For the west coast of North America, as it is in the downstream region of the jet stream, the contribution of this term may be not negligible. It is fine that the authors focus on energy transfer/conversion terms, but the description of the energy flux term need to be more accurate.

To clarify this aspect, we add: Fluxes of energy by the mean flow ($-\nabla \cdot \overline{\boldsymbol{u}}(EAPE + EKE)$) and pressure work ($-\nabla \cdot (\boldsymbol{u}'\Phi')$) are not assessed in this study since they can basically contribute to redistributing energy horizontally, and thus cannot explain the modulations of EAPE and EKE observed in our analysis. Although there are non-negligible local contributions from these terms, mostly associated with the downstream transport of EAPE and EKE by the basic-state westerlies over the North Pacific, their overall contribution is small in comparison to CP and CK when integrated over a large domain.

4. Acknowledgements: I think the reviewers deserve a mentioning here.

We add proper acknowledgments.

**Referee # 2 comments:**

This study is interested in the influence of tropical Pacific SSTs on subseasonal variability in North American SAT during the winter season. The authors use SVD, regression/correlation and composite analyses to investigate how ENSO affects subseasonal variability through modulation of subseasonal eddies - specifically, via changes to the vertical structure of the eddies which have bearing on the amount of baroclinic energy conversion that occurs. The revised manuscript has gone a long way towards addressing a number of concerns about the clarity, interpretation and statistical significance of the study, and I very much appreciate the work the authors have put in. The subject is interesting and relevant to improving our understanding of climate dynamics, as improving near-term climate predictions, and better understanding the large-scale conditions for extreme events. It would be good to see a few final issues resolved before publication.

MAJOR COMMENTS
1. Thanks for the explanation of why this SVD technique was used over traditional ENSO indices. I agree it is worthwhile exploring the "flavour" of tropical Pacific variability associated with SSV of temperature over North America. As I understand, it turns out that the SST pattern from the SVD is ENSO-like, so it doesn't really give us any new information on "flavours" - other than perhaps the discussion of the residual in section 3.6. If so, then it seems okay to leave the title as is, but in other places, this result and implications should be clarified (e.g., ENSO should be replaced by "tropical Pacific variability" in places like the first line of the abstract and title of section 3.2; the abstract and summary should convey that the important thing for North American SSV turns out to be pretty much ENSO). Otherwise, the nice addition/explanation on L105 is completely disconnected from the rest of the manuscript.

Thank you very much for the feedback. Following your recommendations, we have modified the abstract and the title of section 3.2 to make it clear that we assessed the connection between tropical Pacific variability and North-American subseasonal variability and found that ENSO-like variability is dominating this connection.

2. The original review included a comment about the portion of extratropical SSV related to ENSO. I noted that it seemed important to establish this up-front, since later on in Fig. 10, you show an SSV signal unrelated to your ENSO index (SVD1) that is both substantial in amplitude and very similar to the ENSO-related signal. The addition of Fig. 1 showing the climatology is very useful, and I see now about 10% of the local interannual variance is related to ENSO, compared to 50% unrelated. Can you explain a bit more then why you conclude that ENSO plays a prominent role in modulating SSV over North America (L356)? Is the idea that the 10% we get from ENSO is at least predictable, compared to the rest of the SSV that is just internal atmospheric variability? It seems important to mention this in the abstract also.

Thank you for the comment. We now mention in the abstract that $SVD1_{SST}$/ENSO explains a major fraction of the SSV variability over North America that is associated with tropical SST variability. To avoid such misinterpretation that it explains a large fraction of the total SSV variability, we have added: "We find that El Niño-Southern Oscillation (ENSO) explains a dominant fraction of the year-to-year changes in subseasonal SAT variability that are covarying with SST, and thus likely more predictable". We have also added the following discussion in the summary section: "This relationship explains about 77% of the squared interannual covariance between SST and SST-driven SAT variability and a more modest (up to ~10-20% in some sectors) of total subseasonal SAT variability including SST-forced and internal components. Although small, this fraction is nonetheless important because it represents what is predictable from SST variability, unlike atmospheric internal variability which is less predictable."

3. The results would be more compelling if the statistical tests were a bit more systematic, to allow the reader (and the interpretation/discussion) to focus on the robust signals. - some figures use a 95% significance level, others use a 90% level - also, I believe in some cases where "confidence level" is used, it should be "significance level" - some

composites are defined with +/- 1 values of SVD1, others with +/- 0.5 - the correlations in Fig. 6 are quite noisy spatially) and use a rather generous significance level - it seems like field significance should be tested – no significance indicated on Fig. 9, which explores the mechanism.

Thank you very much for the comment. We have standardized our statistical tests to use field significance (Wilks 2016) that we describe in section 2.5 with a significance level of $\alpha_{FDR}=0.1$. We have also standardized composites and eddy structure analyses with a threshold of 0.75, which is midway between the two thresholds that were used previously. In addition, we have added field significance to Fig. 9 to show where regressed geopotential height anomalies are significant. However, we cannot show the significance for heat fluxes since they are a product of two regressed quantities.

4. Section 3.5 is clearer than it was previously. However, I'm not sure the reasoning hangs together with the results as they're presented - it seems some of the arguments would need to be backed up by more rigorous analysis of the temperature distributions and how much they overlap. For example, I don't think we can conclude whether the winter-mean sets the frequency of extremes by shifting the distributions (L322), or whether the presence of a few extreme days determines the winter-mean. Perhaps it would be better to make the discussion more general overall - in some regions it seems the change in SSV broadens the distribution towards one side or the other (cold or warm), and in other regions, while in other regions, we see what may be more a shift. And then show some histograms to bolster the discussion?

We agree with the reviewer's comments. We now avoid implying causality between shifts in the winter mean and the frequency of extremes since winter-means are influenced by extremes. We rather attempt to discuss changes in the probability distribution of temperatures (including the mean and variance) and how they are linked (without implying causality) with changes in the frequency of extremes. Following the reviewer's comment, we have added a new figure (Fig. 11) to show histograms at locations that are affected by shifts and broadenings in the distributions.

OTHER POINTS

- L173: "North America" might be more straightforward than "conterminous..."

Modified

- L173-174: lower-case "northwest-southeast-tilted"

Corrected

- L143 and L179: slightly different filter details for high-pass

This is on purpose. To illustrate the spatial distribution of high-frequency variability associated with migratory cyclones and anticyclones we used a 2-8 day filter to be consistent with previous literature where this band is used frequently. This filter excludes mesoscale systems, tides, and larger synoptic-scale systems at the boundary of subseasonal variability. For the analysis of energetics, however, we used a 10-day high-pass filter to include all variability whose frequency is higher than the subseasonal time scale as defined. This choice does not have any adverse impact on our diagnostics since mesoscale systems and tides do not provide large feedbacks on subseasonal variability. The feedbacks captured are largely due to synoptic systems.

- L231: seems Fig. 6 is mentioned in the text before Fig. 5

We do indeed refer to Fig. 6 before Fig. 5 because it is practical to refer the reader to some information shown in Fig. 6 at that time, but it is more logical to keep the discussion associated with Fig. 6 after the discussion associated with Fig. 5. To avoid confusion, we indicate "(**shown later in** Fig. 6)".

- L239: How was 1500 chosen as the optimal number of resamples?

We found by carrying out repeated trials that significance is stable by this number of sample, thus we chose it to reduce the computational costs. Nonetheless, we increased it to 3000 for the revised manuscript.

- L292: "Differences in the Z500..." - where do we see this?

We now specify that we are referring to Fig. 8 and list Alaska as an example.

- Captions should include more details so that the reader need not go back to the text to look up information, abbreviations, etc.

We have added information about the time scales of variability in Figs. 1, 2, and 12 and indicate the meaning of SSV.

- Fig. 3: colour bar for energy conversion should probably be adjusted

We have adjusted the color bar.

- Fig. 8: nice with the crosses, but they don't show up well in this colour

We change the color to bright green.

References:
Wilks, D. S., 2016: "The Stippling Shows Statistically Significant Grid Points." *Bull. Am. Meteorol. Soc.*, **97**, 2263–2274.

---

## Author Response (AR3)

Dear Dr. Zhang,

We made the following corrections to our manuscript:

1. Line 15-16: Based on the new result in figure 8, the statement on the vertical structure of eddies may need to be revised accordingly.

We write instead: "Structural changes of these eddies are crucial to enhance the efficiency of the energy conversion via amplified downgradient heat fluxes that energize subseasonal eddy thermal anomalies." This statement is supported by our analysis of eddy correlation terms (Fig. 6).

2. Line 295: alco->also

Thank you for notifying us of the typo, we make the correction